# A pictorial account of the human embryonic heart between 3.5 and 8 weeks of development

Jill P. J. M. Hikspoors [1]✉, Nutmethee Kruepunga[1,4], Greet M. C. Mommen[1], S. Eleonore Köhler[1],
Robert H. Anderson [2] & Wouter H. Lamers[1,3]

Heart development is topographically complex and requires visualization to understand its progression. No comprehensive 3-dimensional primer of human cardiac development is currently available. We prepared detailed reconstructions of 12 hearts between 3.5 and 8 weeks post fertilization, using Amira® 3D-reconstruction and Cinema4D®-remodeling software. The models were visualized as calibrated interactive 3D-PDFs. We describe the developmental appearance and subsequent remodeling of 70 different structures incrementally, using sequential segmental analysis. Pictorial timelines of structures highlight age-dependent events, while graphs visualize growth and spiraling of the wall of the heart tube. The basic cardiac layout is established between 3.5 and 4.5 weeks. Septation at the venous pole is completed at 6 weeks. Between 5.5 and 6.5 weeks, as the outflow tract becomes incorporated in the ventricles, the spiraling course of its subaortic and subpulmonary channels is transferred to the intrapericardial arterial trunks. The remodeling of the interventricular foramen is complete at 7 weeks.

[1] Department of Anatomy & Embryology, Maastricht University, Maastricht, The Netherlands. [2] Biosciences Institute, Newcastle University, Newcastle upon Tyne, UK. [3] Tytgat Institute for Liver and Intestinal Research, Academic Medical Center, Amsterdam, The Netherlands. [4] Present address: Department of Anatomy, Faculty of Science, Mahidol University, Bangkok, Thailand. ✉email: jill.hikspoors@maastrichtuniversity.nl

Embryology is a visual discipline. Many aspects of embryonic development are topographically complex, such that three-dimensional (3D) models are exceedingly helpful in fully understanding temporal events. Examples of often used or cited classical models are Born's "Plattenmodellen", and Ziegler's freehand models of embryos, which were studied and described by His[1,2]. Other examples are Blechschmidt's models and drawings of human embryos[3,4], and van Mierop's images of the developing heart[5], which were redrawn by Netter[6]. All these successful approaches have in common that medical artists collaborated with embryologists who had artistic capacities themselves. Because the methods used to make the models were labor-intensive, existing illustrations were often modified rather than new versions being created. An example, documented in detail[7], is the frequently cited treatise of Kramer on the septation of the outflow tract[8]. Such serial modifications, however, tend to propagate concepts rather than observations, and need to be assessed with caution.

The advent of computer-aided reconstruction methods has significantly decreased the time necessary for the reconstruction of sectioned bodies. A recent example is the digital atlas of human development produced by the Amsterdam group[9]. This atlas, however, does not address the development of the heart in any detail. Both qualitative[10] and quantitative[11] tabulations are available for the development of the human heart. Since nomenclature in the developing heart is notoriously variable, a combination of text and illustrations is necessary to provide an understandable account. In this respect, the description of human cardiac development based on magnetic resonance or fluorescent episcopic microscopy is instructive[12]. The spatial resolution and differential staining properties of these techniques, however, are still limited. To our knowledge, no comprehensive primer of cardiac development is presently available that is based on first-hand segmentation of structures of interest identified in histological sections. Our study has visualized such development in human embryos between 3.5 and 8 weeks of development, extending from Carnegie stages 9 through 23. We describe each of the stages, and the features distinguishing them from the previous stage. The evidence can be inspected in the corresponding interactive 3D-pdfs (Supplemental Figs. 1–12). The reader is encouraged simultaneously to read the text and inspect the corresponding interactive PDFs. This is because their rotational options ("live" images) allow a much better understanding of the complex local topography than do "still" images and text.

## Results and discussion
**Distinct developmental features in staged human embryonic hearts**. We describe the conspicuous morphological features of the developing human heart as they appear in consecutive stages of development. The descriptions of the components of the embryonic hearts follow the bloodstream, proceeding from the venous to the arterial pole. The developmental processes in human embryos are, where appropriate, underscored with experimental and molecular data from other mammals, in particular mice, and if fitting, also with data from chickens. To facilitate these interspecies comparisons, Supplemental Fig. 13 shows the relation between developmental stages in human embryos and those in mouse or chicken embryos.

**Carnegie stage 9**. The heart becomes morphologically identifiable when ~26 days have passed since fertilization[13]. The reconstructed specimen is shown in Supplemental Fig. 1. This embryo has developed a neural plate that is flanked by 5 somites and somitomeres. The endoderm, shown in gray, is forming the pharynx. It is continuous at its periphery with the yolk sac, shown in darker gray. The horseshoe-shaped pericardial cavity covers the endoderm in front and laterally, where it becomes gradually narrower to end adjacent to the first somite. Gastrulation begins during CS8 (~23 days of the development;[14]). By CS9, Hensen's node, which localizes gastrulation, is found at the caudal end of the columns of somitomeres.

The heart is located at the cranial margin of the embryo[7,15]. At CS8, midway through gastrulation, bilateral heart fields, also known as cardiogenic plates, form craniolaterally in the embryonic mesoderm[16]. The cranial part of these morphologically still indistinct precursors forms the "first heart field", and differentiates to form cardiomyocytes during or shortly after the formation of the first somite. The remaining, more caudally located and still proliferating precursors, which form the "second heart field", start to differentiate at the transition of CS9 into CS10[17–20]. Cardiomyocyte contractions arise in the lateral regions of the cardiac crescents of mouse embryos as soon as sarcomeric assembly and $Ca^{2+}$ transients can be demonstrated[19,21,22]. Visible contractions develop in mouse embryos when 3 somites have formed[23], while the first cardiac contractions in human embryos can be visualized with ultrasound at days 28 after conception[24]. The first heart field can be visualized in mice by the expression of transcription factors $Tbx5$[25,26] and $Hcn4$[27,28], and the second heart field by the expression of the transcription factor $Isl1$[29]. Fate-mapping studies in mice have shown that the first heart field contributes virtually all cardiomyocytes of the embryonic left ventricle, three-quarters of those in the atrioventricular canal, and half of those making up the atriums and right ventricle. The more slowly evolving second heart field, on the other hand, contributes virtually all cardiomyocytes of the outflow tract, half of the atria and right ventricles, and one-quarter of the atrioventricular canal[18,30]. The systemic venous sinus is a separate part of the second heart field that does not express the early cardiogenic transcription factor $Nkx2$-5, but does express $Tbx18$[31,32]. Based on molecular mouse data, we infer that the center of the heart develops first, and that the upstream venous and downstream arterial components are added successively.

Supplemental Fig. 1, and other reconstructions[9,33] of embryos with ~5 somites, show that the early heart is, like that in mouse embryos[34], bilaterally symmetrical. It consists of two tiny vascular networks that course in front of the foregut, with only few vessels connecting them across the midline. Both vascular networks are surrounded by paired, but partially merged swellings of acellular cardiac jelly. The jelly is enclosed, in turn, within an unpaired, bilaterally symmetrical pericardial cavity with a myogenic visceral wall[15]. The distribution of the jelly, which is produced by the endoderm and the visceral pericardial wall[35], reflects the location of the boundaries of the differentiating left ventricular myocardium. The initially non-luminal endocardial tubes of the cardiac vascular network gradually canalize, but at first contain only few erythrocytes. At the venous pole, the heart tubes are continuous with an extensive venous plexus on the periphery of the endoderm. At the arterial pole, near the buccopharyngeal membrane, the heart tubes pass the pharynx laterally to join the paired dorsal aortas. In front of the heart, the primordium of the transverse septum forms as a shelf of thick mesoderm between the endoderm and the pericardial cavity (Supplemental Fig. 14).

**Carnegie stage 10**. It is within this stage, when ~28 days have passed since fertilization[13], that the heart starts beating[7,23,36]. The reconstructed specimen is shown in Supplemental Fig. 2. The neural plate is flanked by eight somites. It is transforming into a neural tube at the level of somite 4, representing the future junction of the head and neck. The endoderm, shown in gray, is

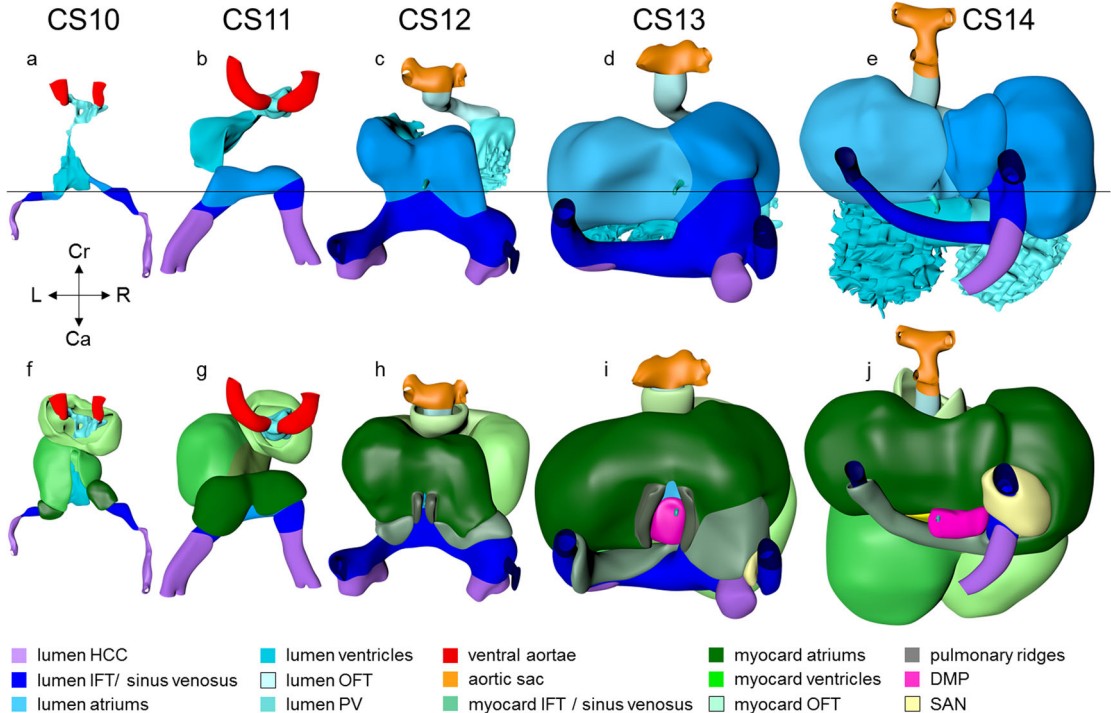

**Fig. 1 Pictorial timeline of the development of the inflow tract.** Dorsal views of embryonic hearts are shown between CS10 and CS14, with panels **a**–**e** emphasizing endocardial continuity and panels **f**–**j** the myocardial coat. The panels are aligned relative to the position of the pulmonary vein (black horizontal line). Note that the distance between the arterial and venous poles of the heart loop does not change during cardiac looping (CS10-CS12;[52,55]). The appearance of a myocardial wall indicates the formation of that compartment in the inflow tract. Myocardium appears in the wall of the atrium at CS10 and in the wall of the systemic venous sinus at CS12. The pulmonary vein, along with its flanking atrial ridges, also begins to form at CS12. The sinus node becomes recognizable as a separate structure at CS13. The left and right atriums are already distinguishable at CS10, but the sinuatrial junction does not become a right-sided structure until CS13. The atrial septum appears at CS14. It is identifiable as the "empty" space between left and right atriums in panel **e**. The hepatocardiac veins are the only source of venous blood for the heart until CS12, when the initially small common cardinal veins appear. All images are also available as preset views in the corresponding 3D-PDFs.

still continuous at its periphery with the yolk sac, which is shown in darker gray.

Within at most 2 days, the heart has transformed into a single endocardial conduit extending between still paired venous and arterial vessels (Fig. 1; see ref. [15]). The single channel has the embryonic left ventricle as its caudal, and the embryonic right ventricle as its cranial component. The umbilical vein, which occupies the boundary of embryonic disk and amnion, and the vitelline plexus, which is situated on the yolk sac, merge at the level of the 4th somite to form the hepatocardiac channels[37]. These channels, in turn, join the systemic venous inflows to the heart. At this stage, both arms of the cardiac inflow tract are transversely oriented vessels, merging in the midline (Fig. 1). This site of union represents the caudal continuity between the first and second heart fields[38], and corresponds with the future atrioventricular junction[39]. It is not yet possible anatomically to identify specific venous tributaries, but the primordiums of the atrial chambers are visible. Cardiac jelly forms a thick cuff around the single endocardial tube, while the outer myocardial wall surrounds the jelly as a cloak, which is open dorsally as the dorsal mesocardium. The dorsal mesocardium connects the heart with the overlying pharyngeal floor, while the transverse septum supports the embryonic ventricle caudally (Fig. 1).

The lumen of the heart resembles that of an hourglass (Figs. 1a–e and 2). At the narrowest part of the hourglass, the dorsal mesocardium has disappeared. At this site, the transverse pericardial sinus, identifiable by the interruption of the dorsal mesocardium in Fig. 1f–j, marks the transition from the descending, or inlet, to the ascending or outlet limb of the

forming cardiac loop. This junction between the embryonic left and right ventricles will eventually be the location of the interventricular foramen. It is at this position, furthermore, that the heart tube bends rightward and, in particular in its cranial part, ventrally (Fig. 2).

The looping of the heart brings out the separation of the second heart field into caudal and cranial portions[40]. In mice, the caudal second heart field gives rise to both atrial chambers at CS10-11, and the systemic venous sinus at CS12 (see refs. [31,41] and Supplemental Table 1). At CS10, the cranial second heart field gives rise to the embryonic right ventricle proximally[42], while the distal portion becomes the myocardial outflow tract at CS11 (see refs. [33,43] and Supplemental Table 1). In mice, the embryonic right ventricle originates from myogenic cells in the second heart field, which also give rise to the muscles of the 1st pharyngeal arch, whereas the outflow tract is covered by cardiomyocytes, which originate in a similar fashion from myogenic cells in the 2nd pharyngeal arch[44,45].

The outflow tract, at the arterial pole, continues extrapericardially as the paired ventral aortas, which extend parallel to the pharyngeal floor in the cranial direction. They then pass perpendicularly to the pharynx, in front of its widening part, which will give rise eventually to the pharyngeal pouches, to join both dorsal aortas. Ventral aortas are found in embryos of all higher vertebrates[46,47], including human embryos during CS10 and CS11. The cranial boundary of the cardiac jelly coincides with the transition of the outflow tract to the ventral aortas. The dorsal aortas course caudally between the dorsolateral wall of the pharynx and the somites, breaking into a plexus where somites

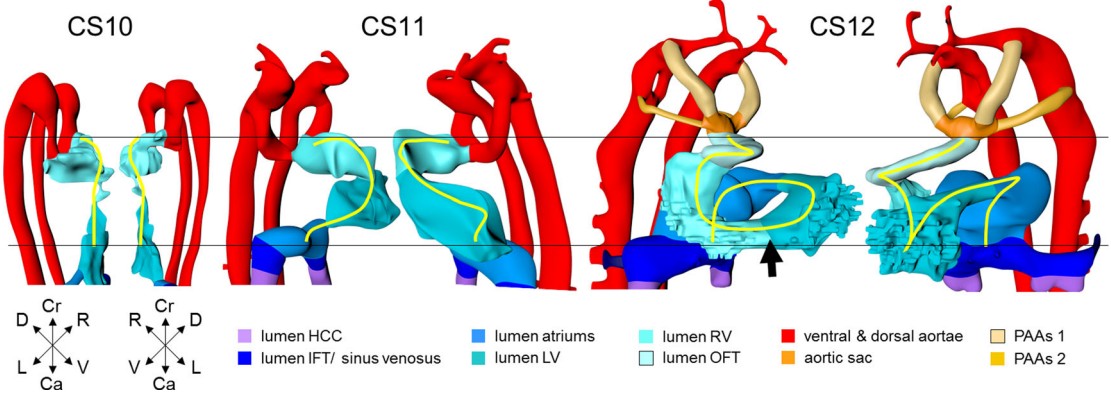

**Fig. 2 Looping of the heart tube.** The panels show ventral right and ventral left views of the cardiac lumen and the adjacent vessels in CS10-12 embryos. The panels were aligned on the arterial and venous poles of the heart loop (black horizontal line). The first signs of looping are seen at CS10, when the dorsal mesocardium disappears at the junction of the embryonic ventricle and outflow tract. The center of the heart tube, represented by the yellow wire, bends rightward and ventrally, in particular in its cranial part. At CS11, the loop extends ventrally due to axial growth and becomes the more pronounced "C" loop. The embryonic left ventricle represents the most ventral portion of the heart loop at this stage. The atrioventricular junction has moved leftward, while the common atrium and distal part of the outflow tract remain midline structures. At CS12, endothelial sprouting into the cardiac jelly marks the boundaries of the ballooning apical parts of the ventricles (arrow). The heart loop between the left ventricle and distal outflow tract further increases in length in a rightward and dorsal direction, with the embryonic right ventricle emerging at its apex (see wire loop). Note that looping has induced two helical twists in the heart axis that meet in the right ventricle. All images are also available as preset views in the corresponding 3D-PDFs.

are forming. By this stage, the roots of the first pair of intersegmental arteries can be recognized between somites #1 and #2.

**Carnegie stage 11**. Human embryos reach the 11th Carnegie stage when ~29 days have passed after conception[13]. The reconstructed specimen is shown in Supplemental Fig. 3. It has developed 13 pairs of somites. The neural plate has partially transformed into a tube, with its neuropores reaching the mesencephalon cranially, and the somitomeres caudally to somite #13, which is equivalent to vertebral level T2. The pharynx by now extends further cranially and has widened, but has not, as yet, given rise to individual pouches. The pericardial cavity extends between the stalk of the yolk sac and transverse septum caudally, the pharynx dorsally, and the forebrain cranially (Supplemental Fig. 14). The entire cardiac tube, except for its caudal non-myocardial inflow tract, is invested in cardiac jelly and has myocardial walls (Fig. 1f–j). The ventricular lumen shows pronounced folds along its longitudinal axis, which represent outward extensions of the endocardial tube that presage the appearance of the ventricular trabeculations in the next stage. At the venous pole, the hepatocardiac channels, formed from the vitelline and umbilical veins, drain into the inflow tract (Fig. 1a–e). Together, the hepatocardiac channels and the inflow tract determine the contour of the cranial intestinal portal. The vitelline veins develop on the craniolateral surface of the yolk sac. Although the cardinal veins have begun to form within the embryo, their connections with the venous pole of the heart have yet to form. Coelomic cells have formed two, still separate proepicardial organs at the junction of the hepatocardiac channel and the inflow tract.

The expansion and the medial fusion of the myocardial walls of the atriums (Fig. 1f–j; see ref. [48]) are indicative of continuing differentiation. The beginning of ballooning of the right atrium, and the leftward transfer of the atrioventricular canal permit the recognition of laterality (Fig. 1; see refs. [40,49]). This laterality involves differences in both lineage[31] and phenotypic properties[41,50] of the right and left atriums. The apical part of the embryonic left ventricle is also beginning to balloon at the outer curvature of the loop. The outflow tract still bifurcates just ventral to the pharynx into the ventral aortas, which continue

dorsally on either side of the pharynx to join both dorsal aortas. The dorsal aortas extend caudally to the dorsal wall of the cloaca and have not yet started to fuse. Near the caudal part of the junction of the midgut and stalk of the yolk sac, an arterial plexus with multiple roots in the ventral wall of the dorsal aorta begins to extend ventrally towards the yolk sac. This plexus represents the earliest stage in the development of the celiacomesenteric trunk. On the left side, one arterial vessel has a markedly bigger diameter than all other components of the arterial network. Slightly more caudal, yet another plexus with multiple roots in the dorsal aortas extends along the cranial wall of the cloaca towards the connecting stalk. This plexus represents the earliest stage in the development of the umbilical arteries. The configuration is remarkably similar to that reported by Tandler in a human embryo with 14 somites[51].

The appearance of the transverse pericardial sinus in CS10 embryos marks the beginning of cardiac looping. The accompanying rightward tilt of the arterial pole, and the leftward tilt of the embryonic left ventricle, are further overt and early signs of asymmetry in these young embryonic hearts (Fig. 2 and Supplemental Fig. 2). In mice at a similar stage of development, growth in the left side of the arterial pole, the ventral side of the loop, and the right side of the venous pole, exceeds that in the corresponding opposite sides[52–54]. These findings suggest that the breaking of symmetry during looping results from the asymmetric distribution of cell-proliferation centers[52,55]. Due to a higher rate of proliferation and myocardial differentiation of mesenchymal cells in the dorsal mesocardium[56,57], and their subsequent insertion into the venous and arterial poles of the heart[33,52,58,59], the length of the limbs of the cardiac loop increases between the left atrium and the arterial pole, in particular in its cranial outlet segment (Fig. 2 and wireframe in Supplemental Fig. 3; see ref. [52]). At the venous and arterial poles, the heart retains its midline connections with the pharyngeal mesenchyme through the remaining parts of the dorsal mesocardium.

**Carnegie stage 12**. Approximately 30 days have now passed since fertilization[13]. The reconstructed specimen is shown in Supplemental Fig. 4. The cranial neuropore has closed, while the caudal neuropore is distal to somite #23, which is equivalent to vertebral

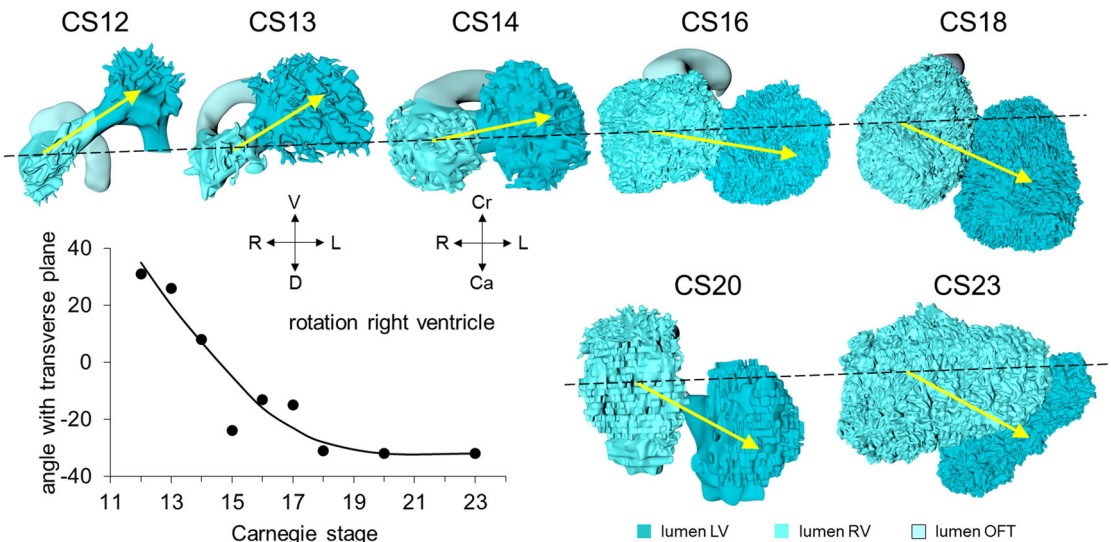

**Fig. 3 Pictorial timeline of the changing position of the developing right ventricle.** The figure shows caudal (CS12 and CS13) or ventral views (CS14-23) of the heart lumen between CS12 and CS23. The difference in the viewing angle reflects the changing curvature of the embryonic axis. The position of the right relative to the left ventricle gradually changes over ~60° between CS12 and CS18 (graph; each dot represents a single embryo). The right ventricle is positioned caudally relative to the left ventricle at CS12 and achieves a more cranial position after CS18. The interventricular foramen is relatively long during CS12-14. The wide space between left and right ventricular lumens after CS20 reflects the appearance of compact myocardium and a thick muscular ventricular septum. The ventricular axes are almost sagittal prior to CS20, and become oblique and leftward at CS23, reflecting the changing shape of the rib cage[212]. All images are also available as preset views in the corresponding 3D-PDFs.

level T12. The fore- and hindgut have further elongated. The first two pharyngeal pouches have formed in the foregut. The buccopharyngeal membrane is breaking up. Surrounded by the pericardial cavity, the heart itself is enclosed on three sides by the transverse septum, the pharynx, and the forebrain (Supplemental Fig. 14). The large systemic veins are still bilaterally symmetrical, and common cardinal veins have formed. Already, the right-sided systemic venous sinus is expanding more rapidly than its left-sided counterpart (Fig. 1). The bilateral proepicardial organs, which originated just cranial and lateral to the inflow tract, have expanded medially over the cranial surface of the transverse septum[60]. A still blind mid-pharyngeal strand, which will eventually canalize to form the common pulmonary vein, can be recognized penetrating the dorsal mesocardium between the arms of the inflow tract (Fig. 1). The atrial margin of the mesocardium is now flanked by paired mesenchymal ridges (Fig. 1f–j). Growth of the right-sided ridge, and its fusion with the primary atrial septum, will eventually commit the common pulmonary vein to the cavity of the left atrium[61–64]. The atrioventricular canal has become recognizable, connecting the left side of the atrial components with the embryonic left ventricle.

Axial growth increases the length of the heart loop, placing the ventricles in a ventral position relative to the atriums, whereas radial growth results in the ballooning of the atrial appendages and apical ventricular components[59,65]. The ballooning was first seen in the right atrium and left ventricle of CS11 embryos (Fig. 1a–e). It results from localized increases in cell proliferation in the outer curvature of the heart loop[5,40,49,66]. The primary myocardium of the embryonic heart is typically bilayered, with a network of thin myocardial strands connecting the layers. The spikes that decorate reconstructions of the lumen of the embryonic left and right ventricles arise from the muscular trabeculations that appear in the ballooning portions of the ventricles between CS12 and CS15 (Figs. 2 and 3). In the embryonic left ventricle, trabeculation of the myocardial wall starts with a few endocardial sprouts penetrating the jelly at CS11. The sprouts increase in number and extend into the inner layer of

the bilayered primary myocardium at CS12. At CS13, the sprouts spread laterally between both myocardial layers, inducing rearrangement of the inner myocardial layer into radial trabeculations. These muscular columns, temporarily covered by "bubbles" of jelly-like extracellular matrix, expand radially during CS14. Their resorption terminates trabecular growth at CS15[67]. The outward and radial growth of the trabeculations leaves intact the contours of the original ventricular endocardial tube[5].

The appearance of endocardial sprouting into the cardiac jelly marks the morphological formation of the embryonic right ventricle at the apex of the heart loop, and in a dorsal position relative to the left ventricle (Fig. 2). The endocardial sprouts also demarcate the left and right boundaries of the interventricular foramen. The size of the cavity, and its trabecular development, are delayed by ~2 Carnegie stages in the embryonic right relative to the embryonic left ventricle (Fig. 3). Differential expression of transcription factors, including *Hand1* and *-2*[68], and *Tbx1* and *-5*[26,40], sustains the differences in growth and shape of the left- and right ventricles. The embryonic right ventricle continues distally into the smooth-walled myocardial outflow tract. Separate left and right bloodstreams now already become visible in the outflow tract[69,70]. Since the inner curvature of the heart does not participate in ballooning and trabeculation, the boundaries of the respective cardiac compartments can only be distinguished along the outer curvature.

When the second heart field starts to contribute cells to the arterial pole of the heart[30,40], the walls of the loop take a helical path between the atrioventricular canal and distal outflow tract (Fig. 2 and wireframe in Supplemental Fig. 4; see refs. [49,71]). This helical configuration can be shown by the expression pattern of the left-sided marker *Pitx2c*[72,73], by lineage tracing[44,49,74] and by the course of the endocardial ridges formed in the outflow tract[7,75]. The spiraling direction is clockwise when looking in downstream direction. Cells from the left side of the caudal second heart field end up cranially (superiorly) in the left atrium and atrioventricular canal, whereas those from the right side end

up caudally (inferiorly)[49]. Similarly, the parietal and septal ridges spiral from the right and left side of the proximal myocardial outflow tract to the left and right side of the distal myocardial outflow tract, respectively. The walls of other structures that form loops, such as the intestines[76], follow strikingly similar courses. During this phase of looping, the elongating muscular outflow tract forms an acute bend between its transversely oriented proximal part, which is also known as the "conus", and its ventrodorsally oriented distal part, also known as the "truncus"[8]. The pronounced "bayonet"[77] or "dog-leg" bend[78] between these parts marks the junction. This bend may be a critical structural element for effective valveless pumping in these early hearts[79]. The presence of the bend permits the outflow tract to be described as having proximal and middle parts, which are myocardial, with the non-myocardial distal part being added when the arterial trunks begin to form in CS15 embryos (Supplemental Table 1).

By this stage, it is possible to recognize the first two pharyngeal arches, along with their accompanying arteries. The vascular space within the ventral pharyngeal mesenchyme that connects the outflow tract with the arteries of the pharyngeal arches is known as the aortic sac. In mice, the endothelium of the first two pharyngeal arches shares its lineage with that of the dorsal and ventral aortas, but differs from that of the subsequent pharyngeal arch arteries[80]. The dorsal aortas have fused between somites #10 and #13 (vertebral levels C6–T2), and continue cranially into the carotid arteries.

**Carnegie stage 13**. This stage, reached at ~32 days after conception[13], is considered "phylotypic". This is because morphologic features and profiles of gene expression are most similar among vertebrate embryos at this stage[14]. The reconstructed specimen is shown in Supplemental Fig. 5. Due to dorsal growth in its sacral region, first noticeable at CS12, the embryonic body axis assumes a helical shape, with the tail region typically on the right side of the body[81]. The heart, within its pericardial cavity, remains surrounded by the transverse septum, the pharynx, and the forebrain. Due to the rapid growth of the brain and foregut between CS9 and CS14, the transverse septum gradually changes in orientation from frontal at CS9 to near-transverse at CS11 (Supplemental Fig. 14). It also "descends" from ~6 somite lengths cranial to the first somite at CS9 to somite #8 at CS13, representing ~3 somites per developmental stage (Supplemental Fig. 14, graph). The large caudal veins remain symmetrical in terms of their size[76], but the vitelline veins have by now been incorporated into the developing liver[37]. The course of the common cardinal veins has changed, following the elongation of the foregut, from being transverse to longitudinal (Supplemental Fig. 14). It is no longer possible to recognize the proepicardial organs, but epicardium is now spreading over the surface of the heart, accumulating in the atrioventricular and interventricular grooves. We show only the thick layer of epicardium in the grooves in our reconstructions.

Myocardium has appeared on the epicardial side of the asymmetrically expanding systemic venous sinus, permitting the definition of the sinus horns (Fig. 1f–j; see ref. [82]). In mice, the myocardial walls of the horns differ from those of the atrial chambers and the pulmonary vein in developing from an *Nkx2-5*-negative, *Tbx18*-positive lineage[31,32,48,50,82]. By this stage, furthermore, the systemic venous sinus drains exclusively into the right side of the atrial chambers through the right-sided sinuatrial junction (Fig. 1a–e). The stem of the solitary pulmonary vein now exits the left atrium through the dorsal mesocardium, but is still blind-ending (Fig. 1f–j; see refs. [62,63]).

Between CS12 and CS13, a subpopulation of endocardial cells undergoes endocardial-to-mesenchymal transformation and colonizes the endocardial jelly[83,84]. This results in the appearance of cellularized endocardial cushions superiorly and inferiorly within the left-sided atrioventricular canal, with the cushions having atrial extensions that encircle the wide interatrial junction. This junction is known as the primary atrial foramen. The myocardial trabeculations remain more advanced in the embryonic left than the right ventricle, while the muscular ventricular septum is no more than a shallow ridge. The cavity of the outflow tract remains surrounded by endocardial jelly, with its smooth-walled myocardial wall extending distally to reach the pericardial reflection. The lumen of the outflow tract then continues via the aortic sac and arteries of the pharyngeal arches to the paired dorsal aortas. There are now three pharyngeal pouches, which interpose between four arches. The 1st pair of pharyngeal arch arteries has disappeared, whereas arteries have formed in the 3rd and 4th arches.

**Carnegie stage 14**. The embryo has now been developing for ~34 days subsequent to fertilization[13]. The reconstructed specimen is shown in Supplemental Fig. 6. Since the cranial somites are no longer identifiable, we revert to spinal ganglia as our reference for the segmental level. The right hepatocardiac channel has become part of the inferior caval vein (Fig. 1). The left hepatocardiac channel is still seen in early specimens of CS14, but has regressed in this advanced CS14 specimen[37]. The confluence of the cranial and caudal cardinal veins has substantially increased in diameter on both sides, while both sinus horns have completely myocardialized. The primordium of the sinus node, with an obvious tail[85], is recognizable as a myocardial cuff at the junction between the right atrium and right common cardinal vein (Fig. 1f–j), which itself is now recognizable as the superior caval vein. In mice, the left-sided marker *Pitx2c* suppresses the development of a sinus node along the left common cardinal vein[50].

The sinuatrial connection, now narrow, is guarded by the venous valves. These valves merge into the spurious septum craniodorsally, and attach in the primary myocardium of the atrial floor caudoventrally. The primary atrial foramen remains surrounded by the atrial extensions of the superior and inferior atrioventricular cushions, with the extension of the superior cushion being a mesenchymal cap on the leading edge of the newly-developing primary atrial septum (Fig. 4; see ref. [86]). The pulmonary vein, which passes between the atrial extensions of the atrioventricular cushions and through the dorsal mesocardium, has now canalized so as to connect with the venous plexuses developing ventral to the lung buds. The rightward margin of the dorsal mesocardium (Figs. 1i–j and 4) is known as the vestibular spine[63] or, more recently, the dorsal mesenchymal protrusion[64]. The atrioventricular canal itself is surrounded by bilayered primary myocardium that extends in the atrial floor to the root of the pulmonary vein and the base of the right venous valve. This extension is also known as the body of the atrium, but shares its lineage and early growth pattern with that of the atrioventricular canal[30,59]. The cushions within the canal now divide its lumen into narrow left and right atrioventricular passages, but have yet to fuse.

The trabeculated free walls of the ventricles continue their ballooning. The left ventricle, thus far made up of cardiomyocytes originating in the first heart field only, also expands by recruiting cells from the atrioventricular canal[87–89]. With the ballooning of the ventricular compartments, it is now possible to recognize the muscular ventricular septum (Fig. 4; see refs. [90,91]). Cell multiplication at its base and a circumferential growth pattern[59,92] bring about a sharp boundary between the left and right sides of the septum[42,93]. Its crest forms the caudal margin of

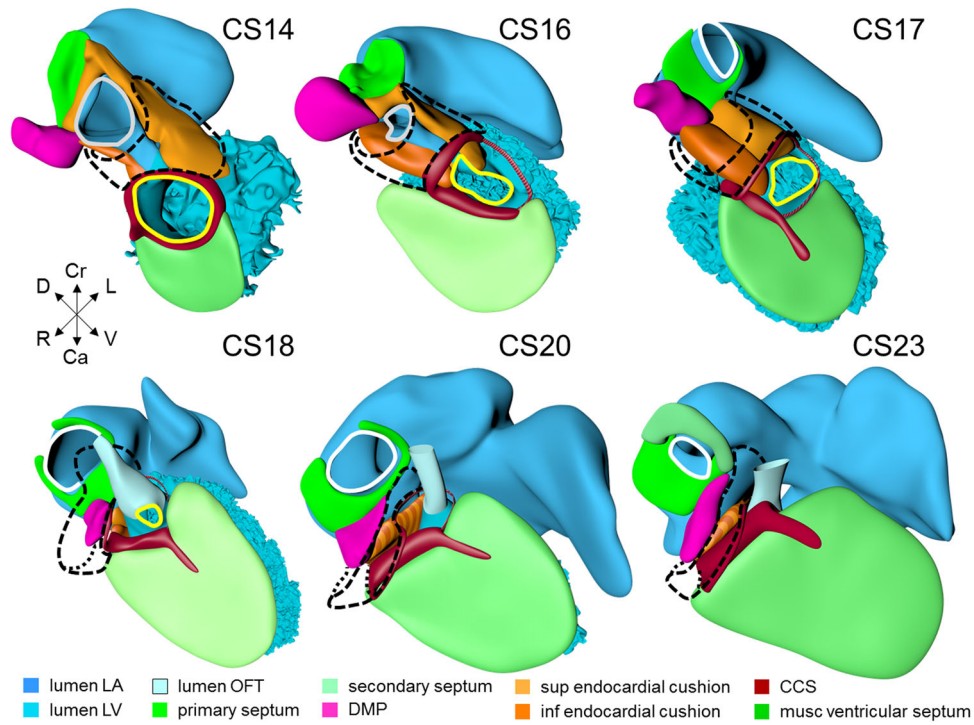

**Fig. 4 Pictorial timeline of the closure of the interatrial & interventricular foramens.** The panels show right ventral views of the left side of the heart. The left atrial and ventricular cavities, the muscular atrial and ventricular septums, the endocardial cushions, the dorsal mesenchymal protrusion, the secondary atrial septum (CS23 only), and the GlN-positive ring bundle are shown. The superior and inferior endocardial cushions, including their atrial extensions, fuse at CS17. This fusion closes the primary atrial foramen and is accompanied by the appearance of a very wide secondary atrial foramen. The dorsal mesenchymal protrusion acquires a position at the base of the atrial septum due to the expansion of surrounding structures. The protrusion muscularizes, along with the mesenchymal cap, starting at CS18, and concomitant with the proximal endocardial ridges of the outflow tract. The borders of the interventricular foramen remodel as revealed by the course of the GlN-positive ring. As soon as septation of the outflow tract is complete at CS18, the myocardialized part of the fused endocardial ridges and the rightward margins of the atrioventricular endocardial cushions combine to decrease the size of the remaining foramen. The closure is complete at CS20. Gray contours: primary atrial foramen; white contours: secondary atrial foramen; yellow contours: interventricular foramen. All images are also available as preset views in the corresponding 3D-PDFs.

the interventricular foramen, with the inner curvature forming the cranial margin (Fig. 4). The myocardium surrounding the interventricular foramen, which is the first component of the second heart field to differentiate[89], can be stained with the "GlN", "Hnk1", or "Leu7" monoclonal antibodies[94]. All of these antibodies recognize a terminal 3-sulfated glucuronic-acid epitope on macromolecules[95]. The very dense appearance of the myocardium of this interventricular ring also makes possible identification of its components in routine histological sections[96,97].

The development of a physical separation between the systemic and pulmonary circulations is known as "septation". In early embryonic hearts of mice[98] and chicken[69,70,99,100], the blood flow is laminar, which limits its mixing. With the appearance of ventricular trabeculations during CS13-14[5,66,101,102], conduction velocity through the myocardial walls increases, and the activation of the ventricle changes from a base-to-apex to an apex-to-base sequence[103–105]. Cardiac pumping, furthermore, switches from a suction, or impedance, to a pulsatile, or piston, mechanism[106]. Because cardiac output increases[107], vortical patterns of streaming[108] and mixing develop, especially downstream of the relatively narrow and still slowly contracting atrioventricular canal and outflow tract[69,100]. The temporal correspondence of the increasing functional effectivity of embryonic hearts[109–111] and anatomical septation, therefore, is not coincidental.

Septation proceeds centripetally from the venous and arterial poles towards the interventricular foramen. Septation of the

inflow tracts becomes feasible once the systemic venous sinus and its tributaries are committed to the developing right atrium, and the pulmonary vein is committed to the developing left atrium. This is seen in CS13 embryos (Fig. 1). The primary atrial septum begins to form at CS14 (Figs. 1a–e and 4), followed by functional septation of the atrioventricular canal by the endocardial cushions into left- and right-sided channels. The borders of the interventricular foramen become remodeled eventually into peri-tricuspid and peri-subaortic portions. These are then separated anatomically by the formation of the membranous septum, which closes the middle portion of the initial foramen at CS20. It is the residual primary myocardium in the inner curvature of the heart that becomes modified during these processes[112]. The expression of GlN in the myocardium surrounding the interventricular foramen facilitates the description of the changes in its shape during the process of septation. Until the end of CS15, however, it remains a flat and round entity, with its borders well described as the primary ring (Figs. 4 and 5).

Septation of the outflow tract proceeds from the aortic sac toward the interventricular foramen. At CS14, the arteries of the 2nd pharyngeal arch have disappeared, while the arteries of the 6th pharyngeal arch have formed. Although only five pharyngeal arches form in amniotes[113], it remains conventional to describe the ultimate arches as being the 6th entities. The pulmonary arteries have yet to appear in this embryo. Pharyngeal arch arteries 3, 4, and 6 form by vasculogenic differentiation[114] of mesodermal cells from the second heart field[80], whereas their smooth muscle coat derives from the cardiac neural crest[115,116].

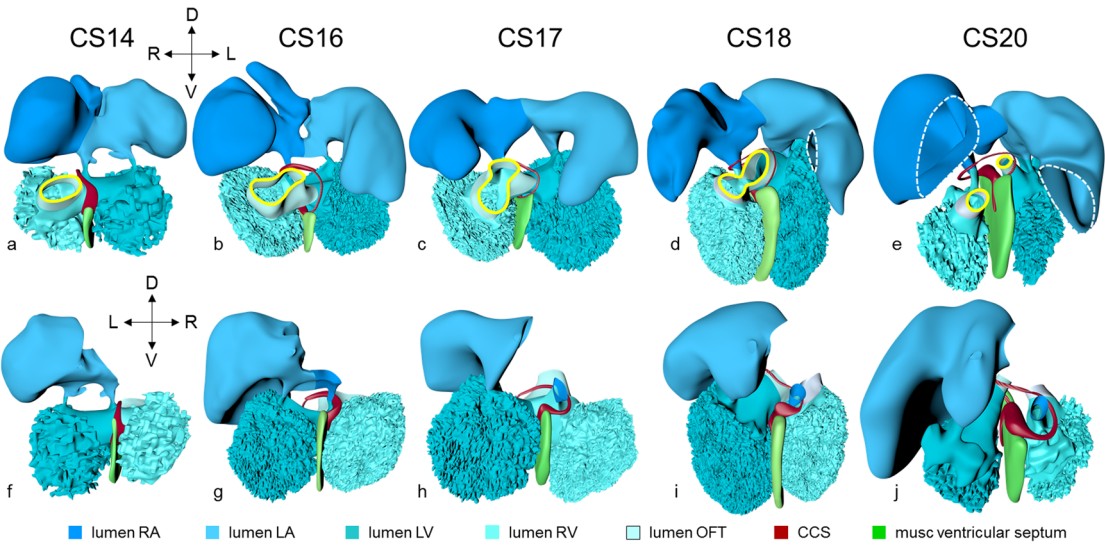

**Fig. 5 Pictorial timeline of the subdivision of the interventricular foramen into the peri-tricuspid inlet, the subaortic outlet, and the membranous septum.** The figure shows the lumens of the atriums, ventricles and outflow tract, and the muscular ventricular septum. Panels **a–e** show a cranial view, and panels **f–j** a caudal view. When first formed, the interventricular foramen is a sagittally oriented interventricular conduit, as visualized by the GIN-positive ring. The craniodorsal part of the foramen, from which the GIN fades away at CS16, is indicated by a thinner, hatched line. The tips of the atrial appendages are clipped in the images for CS18 and CS20 (dashed lines) to permit inspection of the atrioventricular junction and outflow tracts. At CS16, the caudal part of the foramen and GIN ring begin to expand in rightward direction, producing a direct connection between the right atrium and ventricle, which is best seen in panels **f–j**. Meanwhile, the cranial, subaortic part of the foramen, which is best seen in panels **a–e**, gradually expands craniodorsally. Comparing the arrangements at CS18 and CS20, when the septation of the outflow tract is complete, shows that the subaortic, but not the subpulmonary, ventricular outlet is surrounded by the GIN ring. The remaining connection between the right ventricular cavity and the subaortic channel is still present at CS18. It is obliterated at CS20 by the formation of the membranous septum (not itself visible). All images are also available as preset views in the corresponding 3D-PDFs.

In mouse embryos, this part of the neural crest arises between somites levels 1–5, migrates through or around these somites, and through pharyngeal arches 3–6 to the aortic sac[116]. Due to its slow proliferation[58,117], the distal part of the myocardial outflow tract becomes relatively shorter than its proximal part. Up to and including CS13, the distal myocardial boundary reaches the pericardial reflection, with a thick acellular layer of endocardial jelly surrounding the lumen of the outflow tract (Figs. 6 and 7). At CS14, the cells derived from the cardiac neural crest (Figs. 7h–n and 8a–e) and columns of non-myocardial mural cells (Fig. 8a–e) appear as new structures that transform the architecture of the aortic sac and the distal outflow tract.

Cells derived from the neural crest, which surround the arteries of the pharyngeal arches, begin to indent the dorsal wall of the aortic sac. They form a protrusion between its cranial portion, which connects to the arteries of the 3rd and 4th pharyngeal arches, and its caudal portion, which connects to the arteries of the 6th pharyngeal arch. The neural crest cells extend ventrally, having embraced the aortic sac bilaterally, and from there invade the endocardial jelly of the outflow tract as prongs of dense mesenchyme[118]. In this way, they remodel the cuff of endocardial jelly into right- and left-sided columns (Fig. 7a–g; see refs. [118,119]). Meanwhile, endocardial cells that undergo epitheliomesenchymal transformation also populate the endocardial jelly[83,120]. The initially more numerous[75] neural crest cells are necessary for correct positioning of the ridges, and for patterning of the arterial valvar leaflets[119]. The feature, therefore, that distinguishes these ridges from the endocardial cushions of the atrioventricular canal is the presence of neural crest cells. For this reason, we describe the outflow entities as ridges, rather than cushions. The prongs within the ridges take a clockwise-spiraling course when observed in the downstream direction, occupying septal and parietal locations at their junction with the developing right ventricle (Figs. 7h–n and 8a–e; see refs. [7,75]).

The cranial second heart field produces two waves of progenitor cells that are destined to form the outflow tract. The first wave arises at CS10, and contributes to the cranial wall of the muscular outflow tract until CS14 and to the ascending aorta thereafter. The second wave evolves more gradually between CS11 and CS15, and contributes to the caudal wall of the muscular outflow tract and, after CS14, to the pulmonary trunk[121,122]. These cells of the second wave are dorsally continuous with, and probably originate from a phenotypically similar mass of pharyngeal mesenchyme surrounding the trachea[43,123,124]. This "club" of mesenchyme forms during CS13, and remains an identifiable entity during CS14 and CS15 (Fig. 8a–e). The progenitor cells in the club converge and extend into a procession of cells that moves toward, and then into the relatively narrow outflow tract before locally differentiating[122,125,126]. Convergent extension is mediated by the planar cell polarity pathway[127]. When the addition of new cardiomyocytes ceases at CS14, non-myocardial cells start to form the distal portion of the outflow tract. These cells insert themselves cranially and caudally as columns between the remaining myocardial walls[7,128,129]. Consequently, the distal myocardial boundary takes on a fishmouth appearance (Fig. 6d). In contrast to the caudal, or pulmonary, column, which extends to the peritracheal mesenchymal mass, the cranial, or aortic, the column is short when traced into the pharyngeal floor.

In contrast to the neural crest cells, the cells of aortic and pulmonary mural columns do not penetrate the distal endocardial jelly, but maintain an oblique lateral-to medial zone of apposition. Following Tandler[51] and Kramer[8], we will name the endocardial entities they abut "swellings". The cranial, or aortic, swelling differs from the caudal, or pulmonary swelling in that it is invaded by some neural crest cells[119]. The swellings differ from the ridges in that they are initially (CS14 and CS15) confined to a small subsection of the middle portion of the outflow tract near the dog-leg bend (Fig. 7). Consequently, the endocardial jelly, which still surrounds the lumen of the outflow tract as a smooth

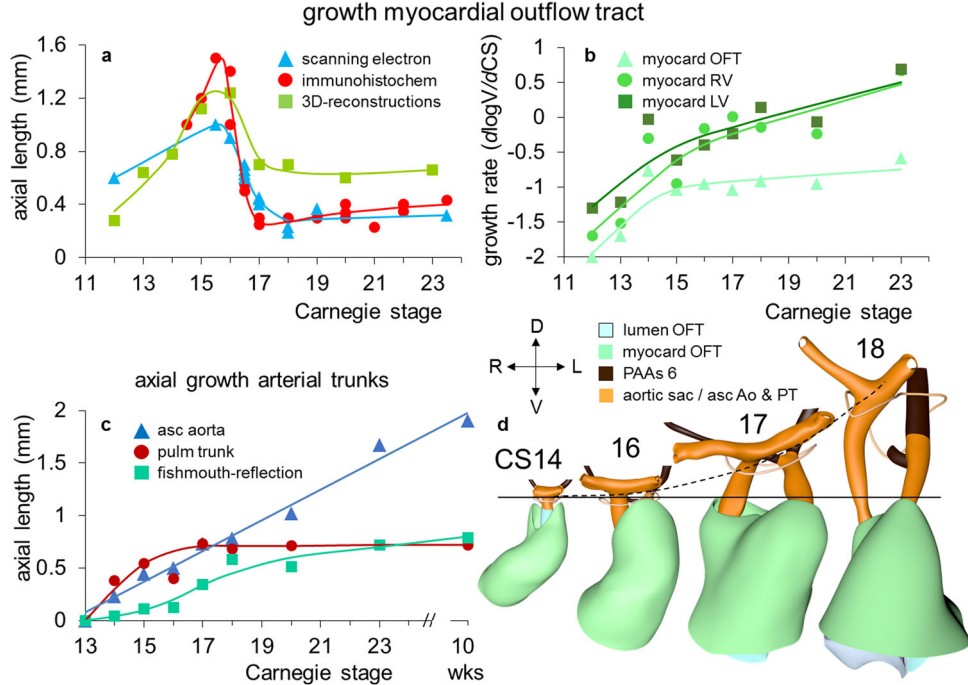

**Fig. 6 Changes in size and shape of the myocardial outflow tract and arterial trunks.** The left-sided graph (**a**) shows the length of the muscular outflow tract between the proximal and distal ends of the endocardial outflow ridges. The green square symbols represent the measurements made in our reconstructions, with the blue triangular symbols taken from measurements made in 14 scanning electron microscopic images[154], and the red circular symbols representing those made in 18 immunohistochemically stained and partially reconstructed hearts[203]. Each dot in the graphs represents a single embryo. There is axial growth of the muscular outflow tract up to CS16, when its length suddenly declines profoundly, with no resumption up to CS23. The right-sided graph (**b**) shows the change in volumetric growth rate of the myocardium of the outflow tract (light green triangles), right ventricle (green circles), and left ventricle (dark green squares). The growth rate in all three compartments declines at CS15, with the growth of the outflow tract being practically nonexistent. The lower left graph (**c**) shows the axial growth of the arterial trunks. The ascending aorta (blue triangles) increases continuously in length between CS14 and 10 weeks of development, whereas axial growth of the pulmonary trunk (red circles) stops after CS17. The distance between the distal myocardial border and the pericardial reflection (green squares) increases little to CS16, indicating that the myocardial jaws of the fishmouth stay close to the pericardial reflection, whereas the myocardial border moves away from the reflection concomitant with the abrupt shortening of the myocardial outflow tract after CS16. Panel **d** shows cranial views of the myocardial outflow tract, the arterial trunks, and the pericardial reflection (wire loop) to visualize the axial growth of the aortic trunk (dashed line). The images are aligned to the distal myocardial border of the outflow tract (black line), with the fishmouth representing the lateral indentations of the distal myocardium between CS14 and CS16. The slits in the jaws of the myocardial fishmouth are occupied by the non-myocardial mural columns (see Fig. 8a–e). The 6th arch arteries (dark brown) stem from the pulmonary trunk, while the ascending aorta is recognizable by its lateral "horns". All images are also available as preset views in the corresponding 3D-PDFs.

cuff at CS13, reorganizes distally into four orthogonal columns, while only two columns persist proximally (Fig. 7a–g).

**Carnegie stage 15**. At this stage, ~36 days have passed since fertilization[13]. The reconstructed specimen, although one of the best CS15 specimens of this stage in the Carnegie collection (http://virtualhumanembryo.lsuhsc.edu/demos/Stage15/Intro_pg/Intro.htm), suffered from venous congestion. To decide whether or not this condition was atypical, we surveyed the Blechschmidt collection of human embryos. Supplemental Fig. 15 shows that 25% of the 32 embryos that had been graded as good or excellent suffered from a moderate to severe degree of venous congestion. This finding suggests that cardiac failure is a common cause of death in human embryos in the 5th and 6th weeks of their development. Compared to the embryo shown for CS14, the changes in the arrangement of the systemic veins and venous sinus are limited, except that a small left hepatocardiac channel was still present in this relatively less advanced CS15 embryo[130]. The left hepatocardiac channel was still discernable in all four CS14 embryos, and in one of the six CS15 embryos of the Blechschmidt collection. The latter embryo did not suffer from venous congestion. The configuration of the atrial chambers and

their topographic relation with the pulmonary vein are similar to those in the previous stage[61–63]. The breakdown of the dorsal portion of the primary atrial septum has created, however, a secondary atrial foramen. In the Blechschmidt collection, we found two CS15 embryos who had not yet formed a secondary foramen and two others who had, like the reconstructed specimen, both primary and secondary foramens. Accordingly, McBride et al.[11] reported that ~50% of the CS15 embryos in the Carnegie collection had developed a secondary foramen. We found no notable changes in the architecture of the ventricles. The GlN-positive interventricular ring is still a planar structure[131], but a widening of the crest of the muscular ventricular septum identifies the developing branching component of the atrioventricular conduction axis[132].

CS15 is the most advanced stage in which the arteries within the pharyngeal arches retain their symmetry, albeit that the portions of both dorsal aortas between the arteries of the 3rd and 4th arches, known as the carotid ducts, have markedly decreased in diameter[133]. By this stage, the arteries of the left and right 6th pharyngeal arches have each given rise to a pulmonary artery, which extends caudally within the pharyngeal mesenchyme along the trachea. The most pronounced developmental changes are to be seen in the arrangement of the middle portion of the outflow

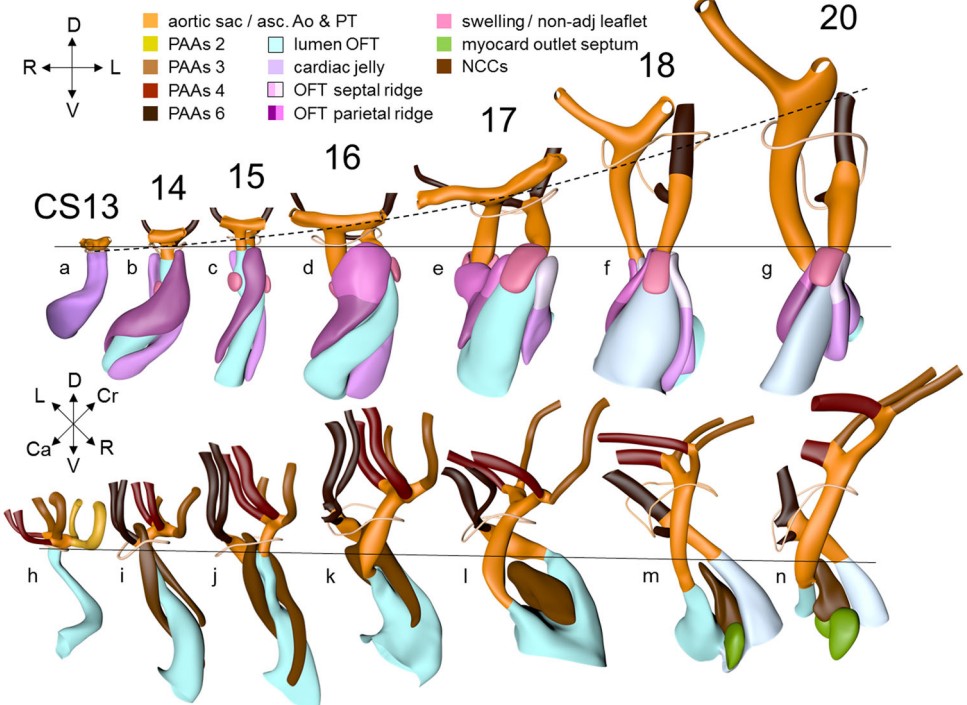

**Fig. 7 Pictorial timeline of the changes in size and shape of the lumen, endocardial ridges and swellings, and neural crest prongs of the outflow tract.**
The images are aligned on the location of the developing pulmonary valve (black line). Panels **a**–**g** show cranial views of the outflow-tract lumen flanked by the parietal and septal endocardial ridges, and aortic and pulmonary swellings. The viewing angle is the same as for Fig. 6d. The arterial trunks are shown for identification of the subaortic and subpulmonary channels. The ascending aorta is recognizable by its lateral horns. The widening of the distal portions of both outflow ridges during CS16 and CS17 presages their allocation to the right or left semilunar leaflets of the aortic and pulmonary arterial valves at CS18. The valvar portions are marked by a less dark tint of the coding color, and are confined to the distal portion of the myocardial outflow tract. The dashed black line in panels **a**–**g** show the axial growth of the intrapericardial component of the aortic trunks. The saddle-shaped wire loop shows the position of the pericardial reflection. Panels **h**–**n** show the lumen of the outflow tract as seen from the right side, showing the neural crest cells within the aortopulmonary septum extending as columns of dense mesenchyme into both endocardial outflow ridges. The fusion of these columns creates a temporary "whorl" of neural crest cells between the subaortic and subpulmonary channels. The neural crest cells largely disappear between CS18 and CS23 due to intense apoptosis[166], with invading cardiomyocytes simultaneously populating the shell of the septum[75,213]. All images are also available as preset views in the corresponding 3D-PDFs.

tract and the aortic sac. Continued axial growth within the myocardial part of the outflow tract (Fig. 6a) has all but eliminated the dog-leg bend. The aortopulmonary septum, initially seen at CS14 as a transverse protrusion extending from the dorsal wall of the aortic sac between the origins of the arteries of the 4th and 6th pair of pharyngeal arches, now extends obliquely in a ventral direction toward the distal margins of the endocardial ridges in the middle portion of the outflow tract (Figs. 7 and 8a–e; see ref. [134]). In consequence, the aortic sac acquires a dextrocranial systemic component, which connects the subaortic part of the outflow tract with the arteries of the 3rd and 4th pharyngeal arches, and a sinistrocaudal pulmonary component, which connects the subpulmonary part of the outflow tract with the arteries of the 6th arches (Figs. 6–8). The intrapericardial part of the systemic component can be labeled with Mef2c-Cre[135], an often-used marker of the cranial second heart field, and becomes the ascending aorta. The lateral horns of the aortic sac remain unlabeled, and become the extrapericardial part of the ascending aorta, the brachiocephalic trunk, and the initial part of the transverse aortic arch. The pulmonary component becomes the pulmonary trunk, an entirely intrapericardial vessel. It is the ventral growth of the aortopulmonary septum, therefore, which initiates the anatomical separation of the arterial pole of the heart[128,136], along with the formation of the non-myocardial distal portion of the outflow tract. The configuration of the distal myocardial jaws and the interposed mural columns remains unchanged relative to that in CS14. Accordingly, the distal myocardial jaws still extend close to the pericardial reflection (Fig. 6c, d). As in the CS14 embryos, tissue with the phenotypic property of the pulmonary mural column extends dorsally to the dense mesenchyme that surrounds the trachea (Fig. 8a, b).

The prongs of neural crest cells, which dorsally are continuous with the neural crest cells in the pharyngeal floor and the aortopulmonary septum[118], can now be traced ventrally into the proximal outflow ridges. By this stage, it becomes possible to recognize the sites of formation of the arterial valves as increasingly narrow passages between the endocardial ridges medially and the swellings laterally[8]. These passages are recognizable histologically by their lining with intensely staining, cobble stone-shaped endocardium. Separate aortic and pulmonary channels are now identifiable in luminal casts of the middle portion of the outflow tract. They extend from the initial site of the dog-leg bend to the distal boundary of the myocardium of the outflow tract (Fig. 8f–j). Until fusion of both endocardial ridges occurs during CS17, the subaortic and subpulmonary channels remain connected by an aortopulmonary foramen, which is bounded dorsally by the leading edge of the aortopulmonary septum. At this distal location, the parietal and septal ridges occupy craniosinistral and caudodextral positions, respectively, with the still small swellings occupying the spaces in between (Fig. 8a–e).

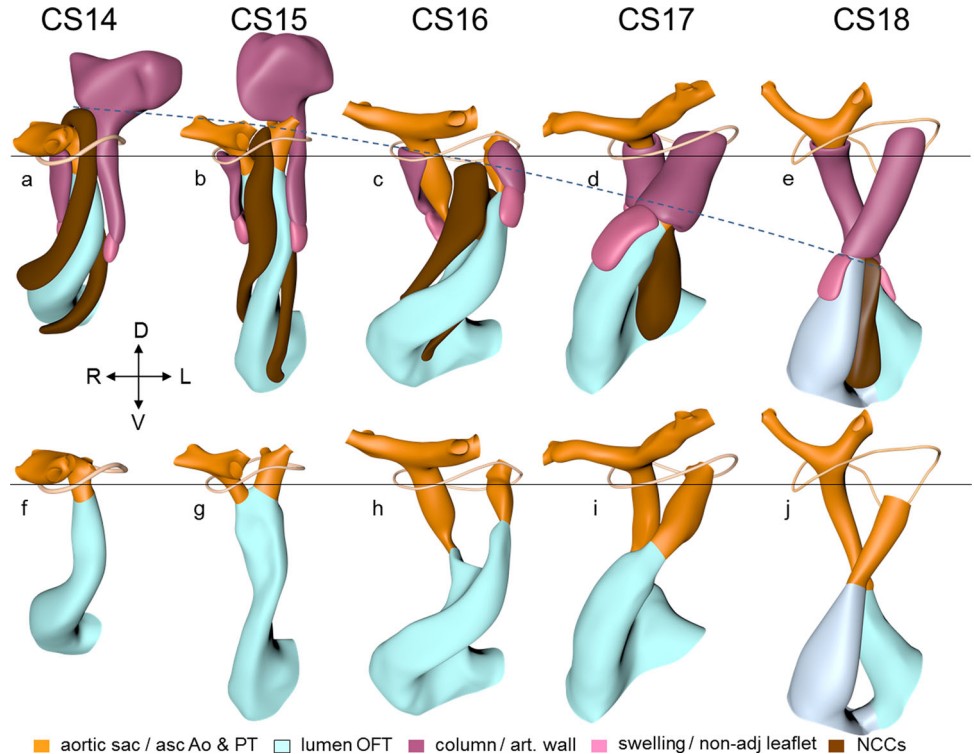

CS14　CS15　CS16　CS17　CS18

■ aortic sac / asc Ao & PT　□ lumen OFT　■ column / art. wall　■ swelling / non-adj leaflet　■ NCCs

**Fig. 8 Pictorial timeline of the appearance of the non-myocardial walls of the arterial trunks.** The panels are aligned on the pericardial reflection, shown by the wire loops, as in Fig. 7. Panels **a**–**e** show the lumen of the outflow tract, with the arterial trunks, the columns of neural crest, and the non-myocardial mural columns. The ascending aorta is recognizable by its lateral horns. The neural crest cells and intercalating non-myocardial tissues invade the distal wall of the outflow tract during CS14. The mural cells are first seen as relatively short aortic or cranial, and pulmonary or caudal columns. During CS14 and CS15, the pulmonary column is continuous dorsally with a club-like condensation of peritracheal mesenchyme, which has disappeared at CS16. The endocardial swellings associated with the aortic and pulmonary mural columns are relatively small during CS14 and CS15, but increase in size from CS16 onwards to begin their transformation into the dorsal and ventral semilunar leaflets, respectively, at CS18. As shown by the interrupted line, there is a gradual increase in the distance between the pericardial reflection and the plane of the valvar primordiums (*cf.* Fig. 7). Panels **f**–**j** show the same view of the lumens. Note that the prongs of neural crest mesenchyme mold the subaortic and subpulmonary channels during CS15 and CS16. The fusion of these prongs into a central whorl marks the separation of the subaortic and subpulmonary channels during CS17 and CS18 (the separation of these channels is shown from a different perspective in Fig. 9b). All images are also available as preset views in the corresponding 3D-PDFs.

**Carnegie stage 16**. This stage is reached at ~38 days after conception[13]. The reconstructed embryo is shown in Supplemental Fig. 8. The systemic venous sinus, its sinuatrial valves, and the sinus node have largely retained the appearances seen in the previous stage. The pulmonary vein remains a solitary and narrow channel. The primary atrial septum, with its mesenchymal cap, has extended further toward the atrioventricular canal (Fig. 4). This reduces the size of the primary atrial foramen, but the atrioventricular cushions still have to fuse and a secondary atrial foramen still has to form in this embryo. This arrangement is seen in two CS16 embryos of the Blechschmidt collection, while a secondary atrial foramen has already formed in another embryo. In a 4th embryo, the atrioventricular cushions have fused, which brings about the closure of the primary atrial septum. This range in developmental progress is in line with that reported by McBride et al.[11] The expansion of the atrial chambers to either side of the outflow tract reveals pronounced growth of the atrial appendages (Fig. 5a–e). With continuing caudal expansion of the atrium, the right border of the dorsal mesenchymal protrusion expands, like a spine, into the atrial cavity, growing between the atrial surfaces of the superior and inferior atrioventricular endocardial cushions (Fig. 4; see refs. [137,138]).

The embryonic left and right ventricles are now of similar size and occupy a transverse plane (Fig. 3). The changing boundaries of the interventricular foramen can still be followed conveniently in hearts stained for the GlN antigen (Figs. 4 and 5). In the inner curvature of the heart, the right wall of the atrioventricular canal continues into the caudal part of the interventricular foramen. Rightward expansion of the confluent part of these structures across the muscular ventricular septum has produced a direct connection between the right atrium and the right ventricle (Fig. 5f–j; see ref. [139]). Subsequently, the cranial portion of the interventricular foramen will evolve into the channel between the left ventricle and the subaortic outlet (Fig. 5a–e). At this latter location, however, the primary ring (hatched section) has lost its GlN expression[94], but remains identifiable as part of the central conduction system in birds[140]. In the reconstructions, we have presumed that it lies, as in birds, in the inner curvature at the junction of the left ventricle with the outflow tract.

The length of the myocardial portion of the outflow tract increases approximately fourfold in the 8 days between CS12, when it can be first differentiated from the embryonic right ventricle, and CS16 (Fig. 6a), underscoring its continuous axial growth. The distal tongues of the myocardial outflow tract still extend close to the pericardial reflection (Fig. 6c, d), but relative to the diameter of the outflow tract, their length declines. Similarly, the aortic and pulmonary mural columns become relatively shorter. The ascending aorta and pulmonary trunk have increased substantially in length since their appearance at CS14, but axial growth of the pulmonary trunk ceases after CS16 (Fig. 6c). This cessation of growth coincides with, and may reflect,

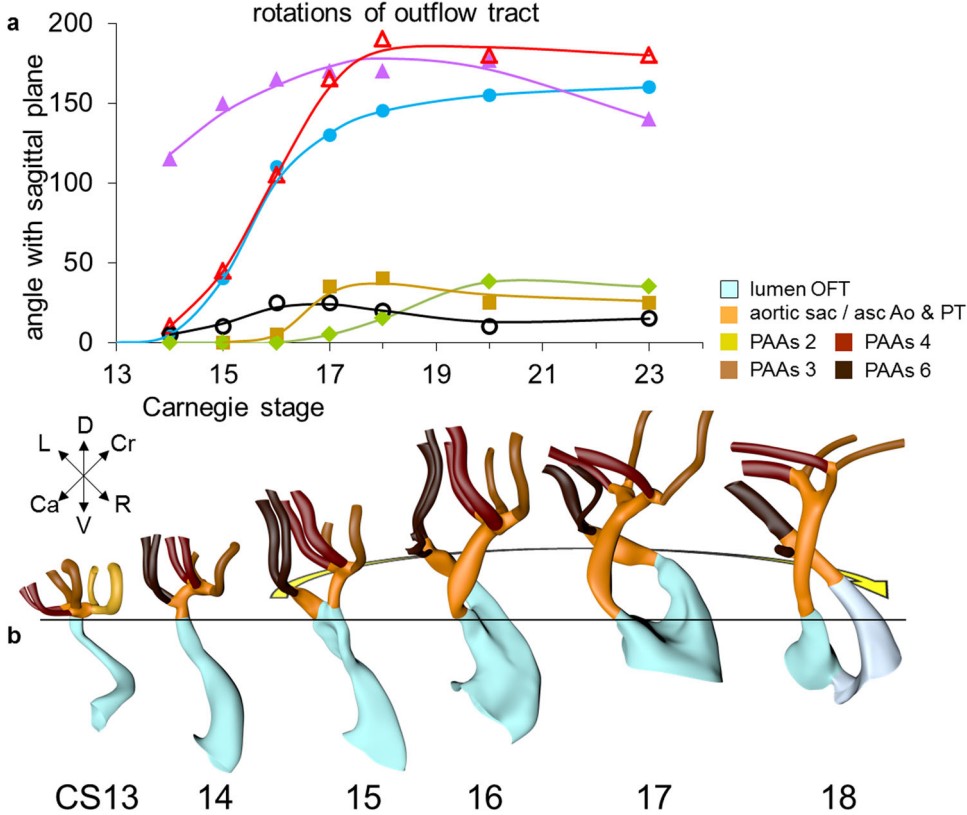

**Fig. 9 Changes in the course of the (sub-)aortic and (sub-)pulmonary channels.** The graph (**a**) shows the changes in the degree of spiraling of the bloodstreams in time and place. The reference plane is sagittal and each dot represents a single embryo. Axial changes in the position of the components of the outflow tract were measured as described in Supplemental Fig. 18. The line connecting the center of the ridges at their proximal ends is shown in purple solid triangular symbols, with the comparable line at their distal ends shown in red open triangular symbols. The blue solid circular and black open circular symbols identify the line connecting the centers of the orifices of the ascending aorta and pulmonary trunk at their proximal and distal ends, respectively, with the distal end of the ascending aorta measured at the pericardial reflection. The brown square symbols show the movement of the distal endocardial ridges relative to the distal orifices of the subaortic and subpulmonary channels. The green diamond-shaped symbols show the asymmetric development of the horns of the aortic trunk measured as the angle of the lines connecting their junctions with the pharyngeal arch arteries. Comparison of the red (open triangular) and purple (solid triangular) symbols shows that, by CS17, the endocardial ridges have lost the initial spiraling arrangement identifiable at CS14. The change in orientation of the myocardialized proximal ridges between CS20 and CS23, as they transform into the subpulmonary infundibulum, accounts for the decline in the purple symbols (see Fig. 10 for morphological details). The compensatory spiraling of the intrapericardial course of the arterial trunks, as shown by the blue symbols, reflects the oblique ventral extension of the aortopulmonary septum, with the black symbols showing that the change in position of the arterial trunks at their connection with the pharyngeal arch arteries contributes to a much lesser extent. The images in panel **b** (same viewing angle as Fig. 7h–n) are aligned on the location of the developing aortic valve (black line). The horizontal yellow arrow shows the changing position of the (sub-)aortic and (sub-)pulmonary channels in the middle and distal portions of the outflow tract. These channels separate between CS14 and CS15 in the distal outflow tract, during CS16 and CS17 in the middle outflow tract, and between CS18 and CS20 in the proximal outflow tract. All images are also available as preset views in the corresponding 3D-PDFs.

the depletion of the peritracheal cell mass ("club") that is present at CS14 and CS15 (Fig. 8a–e). At this stage of development, separate flows of blood reach the systemic and pulmonary arch arteries[7,11], but the endocardial outflow ridges have still to fuse mutually, and with the aortopulmonary septum. Hence, a narrow aortopulmonary foramen is still present distally between the subpulmonary and subaortic channels. The connections of the subaortic and subpulmonary channels with the ascending aorta and pulmonary trunk, respectively, now occupy left and right positions (Fig. 8; see refs. [74,133,141]). This increasingly spiraling course of the intrapericardial arterial trunks corresponds in time with the unwinding of the spiraling course of the muscular outflow tract and its endocardial ridges[142] (Figs. 7 and 9). The still short swellings, which guard the narrow lumen of the developing arterial valves laterally, follow the unwinding course of the main endocardial ridges. Meanwhile, the distal part of the endocardial ridges of the outflow tract begins to increase in diameter relative to the proximal counterparts (Fig. 7a–g). This

increase in size presages the remodeling of their distal surfaces into the arterial valvar leaflets during the next two stages. The carotid ducts have narrowed further. The artery of the right 6th arch is now narrower than the left one, in particular just distal to the origin of the right pulmonary artery. The diameter and perfusion of the left-sided dorsal aorta further increase[143].

**Carnegie stage 17.** At this stage, ~40 days have passed since fertilization[13]. The reconstructed specimen is shown in Supplemental Fig. 9. Compared to preceding stages, limited changes were noted in the arrangement of the systemic venous sinus. The superior and inferior atrioventricular cushions, including their atrial extensions, have fused in nearly all embryos at this stage, thus closing the primary atrial foramen (Fig. 4; see ref. [11]). Accordingly, a wide secondary foramen has formed due to the breakdown of the dorsal portion of the primary atrial septum in these hearts. The site of fusion of the atrioventricular cushions in the atrium is reinforced on its right side by the dorsal

mesenchymal protrusion, or vestibular spine. Myocardium surrounds the stem of the pulmonary vein, which from this stage onwards begins to expand radially, suggesting an increase in blood flow (Supplemental Fig. 16). Small lateral cushions have appeared in the left and right margins of the atrioventricular canal.

The well-developed trabeculations of left and right ventricles form a complex 3-dimensional network[66], and have developed extensive gap-junctional contacts[144,145] and myofibers[146]. These properties reflect their faster conduction and stronger contraction[147,148]. The appearance of a compact left ventricular myocardial wall shows that the multiplication of cardiomyocytes at the epicardial side of the ventricular walls now exceeds that in the inner trabecular layer[5]. The compact myocardium does not arise, as is often suggested, by condensation of the trabecular network[149]. While the muscular ventricular septum develops equally from right- and left ventricular contributions during the phase of trabecular growth, the contribution of cardiomyocytes now becomes proportional to growth in the compact ventricular walls[92]. The peri-tricuspid part of the interventricular foramen, encircling the right atrioventricular orifice, continues to expand in a rightward direction, while its subaortic part remains bordered cranially by the inner curvature and caudally by the crest of the muscular ventricular septum (Fig. 5).

Between CS15 and CS20, the volume of the myocardium of the outflow tract hardly increases in both rodent[58,150] and human embryos (Fig. 6b; see ref. [151]). In overall architecture, nonetheless, it undergoes an impressive change in appearance. Its length decreases abruptly to <50% of its original length in the ~2 days separating CS16 from CS17 (Fig. 6a; see ref. [152]). Much of the effective shortening can be attributed to it changing from a long tubular configuration at CS16 to a shorter figure-of-eight configuration at CS17 (Fig. 6d). The waist of the figure of eight corresponds with the developing medial walls of the subpulmonary and subaortic channels. In addition, the proximal portion of the outflow tract itself becomes progressively more wedge-shaped, with its greatest length along the subpulmonary channel and its shortest length along the subaortic channel. The proximal portion of the outflow tract of birds undergoes an almost identical change in appearance at a comparable stage of heart development[153]. The changing shape reflects the ongoing incorporation of the smooth-walled proximal outflow tract into the right ventricle as its infundibulum, and into the left ventricle as its aortic vestibule[26,152,154–156].

The aortopulmonary septum[157] has fused with the septal and parietal ridges. The fusion of the endocardial ridges mutually now has passed through the middle portion of the outflow tract. As a result, the neural crest cells of the aortopulmonary septum and the prongs become sequestered inside the ridges as a dense central "whorl" of cells between the anlagen of the arterial valves (Figs. 7h–n and 8a–e). The developing aortic and pulmonary roots themselves occupy right-caudal and left-cranial positions, respectively (Fig. 8f–j), which represents their definitive position (Fig. 9a). The distal parts of the septal and parietal ridges undergo a similar change in shape as the corresponding part of the myocardial outflow tract. They now each resemble a heart-shaped structure with wide downstream "ears" separated by a median furrow that presages its separation and allocation to the left- and right semilunar leaflets of the aortic and pulmonary roots, respectively (Fig. 7a–g). The lateral swellings have also increased markedly in size at this stage (Fig. 8a–e). Concomitantly with the distal widening of the ridges and swellings, there is movement in their position relative to the distal orifices of the subaortic and subpulmonary channels (Fig. 9a). The ridges in the proximal outflow tract have still to fuse.

Since the myocardium of the outflow tract hardly proliferates (Fig. 6b; see refs. [58,150,151]), and the addition of new

cardiomyocytes from the second heart field has ceased, the fishmouth has lost its characteristic appearance. The non-myocardial cells that continue to populate the distal outflow tract, now interpose between the distal margin of the myocardial walls and the pericardial reflection. Consequently, the myocardial boundary moves away from the pericardial reflection.

Concomitant with the marked reduction in length of the myocardial outflow tract, the pulmonary trunk ceases to grow. The ascending aorta, in contrast, continues to increase in length well beyond CS23 (Fig. 6c). The inner layer of the smooth muscular wall of both arterial trunks derives from the neural crest, and the outer layer from the second heart field. However, the wall of the ascending aorta is mainly derived from the neural crest[121,135,158], whereas the wall of the pulmonary trunk originates predominantly in the second heart field[121,122,124]. These different contributions may well explain the different growth characteristics of the vessels. The epicardium that covers the intrapericardial arterial trunks expresses the morphological and molecular features of the nearby pericardium. It consists of a sheet of densely packed cuboidal cells rather than of squamous cells, which characterize the epicardium. It also expresses genes that characterize the pericardium[159–161]. This "arterial" epicardium seems to form locally, since it spreads across the arterial pole even after the removal of the proepicardial body[162]. Collectively, these data indicate that the arterial pole of the heart derives from the ventral wall of the pharynx, with the non-myocardial tissues entering the walls of the distal part of the intrapericardial outflow tract to form its arterial component.

Extensive changes have occurred at the arterial pole. Both dorsal aortas are still present, but the left- and right-carotid ducts have disappeared. Marked narrowing is now seen in the diameter of the right-sided dorsal aorta between the take-off of the 7th cervical segmental artery and its confluence with the left-sided dorsal aorta, while the lumen of the artery of the right 6th arch has all but obliterated distal to the origin of the right pulmonary artery.

**Carnegie stage 18**. At this stage, ~43 days have passed since fertilization[13]. The reconstructed specimen is shown in Supplemental Fig. 10. Compared to the previous stage, the venous part of the heart has only changed to a limited extent. Pectinate muscles, first identifiable as small stubs of the inner myocardial layer at CS14, now begin to expand in both appendages. The primary atrial septum and the secondary foramen remain comparable to the previous stage (Fig. 4). Cardiomyocytes are now populating the dorsal mesenchymal protrusion, or vestibular spine, and the mesenchymal cap to form the well-developed ventro-caudal muscular rim of the atrial septum[137]. The diameter of the pulmonary veins has increased further (Supplemental Fig. 16b). Reflecting atrial growth, the myocardial atrioventricular canal changes in appearance from tubular to funnel-shaped, becoming transformed into the vestibules of the atrioventricular valves. The position and relative size of both lateral endocardial cushions in the canal do not change.

While the structural components that are responsible for ventricular septation become morphologically identifiable, the right ventricle gradually begins to occupy a more cranial position relative to the left ventricle (Fig. 3). The junction of the left ventricular inlet and outlet, with the latter still represented by the peri-subaortic component of the interventricular foramen, is guarded mainly by the superior endocardial cushion (Fig. 4). The lesser curvature, forming the cranial margin of the foramen, is still muscular. The junction between the right ventricular inlet, known as the tricuspid "gully"[139], and the outlet is formed by the fusing superior and right-lateral endocardial cushions. By this

stage, only a small connection, representing the middle part of the original interventricular foramen and also known as the tertiary interventricular foramen[163], remains between the cavity of the right ventricle and that of the root of the subaortic outflow tract. This middle part of the interventricular foramen is bounded ventromedially by the muscular ventricular septum, dorsolaterally by the fused proximal ridges of the outflow tract, and dorsally by the fusing rightward margins of the atrioventricular cushions (Figs. 4 and 5, and next paragraph). The location of the borders of the right ventricular (tricuspid) inlet and left ventricular (subaortic) outlet parts of the original interventricular foramen can still be visualized by the shape of the GlN ring (Fig. 5). Because its subaortic portion no longer expresses the epitope, we have assumed its position to be in the inner curvature, where it is found in embryonic chickens[140,164]. The GlN-positive tissue in the septal structures identifies the location of the atrioventricular conduction system[94,165]. In contrast to the relatively narrow atrioventricular junctions, the junctions between the ventricles and the bases of the subaortic and subpulmonary parts of the outflow tract are wide. The subaortic component of the proximal outflow tract, however, still remains positioned above the cavity of the right ventricle, but its cavity is now contiguous with that of the left ventricular outlet. This remodeling provides the left ventricle with unhindered vascular access to the subaortic outflow channel, thus allowing closure of the middle portion of the interventricular foramen to proceed, with closure completed at CS20.

The wedge shape of the myocardial component of the outflow tract, with its blunt side over the outer curvature, and the sharp edge in the inner curvature, becomes more pronounced. The septal and parietal endocardial ridges have now fused across their entire length (Figs. 7h–n, 8, and 9b). The distal-to-proximal fusion of the ridges occurs exclusively in the endothelium overlying the prongs of neural crest cells. The site of fusion is, therefore, marked by the dense central whorl of neural crest cells (Figs. 7h–n and 8a–e). The whorl subsequently disintegrates rapidly due to apoptosis[166–169], but some neural crest cells persist in the semilunar valves[119,170], with other remnants sometimes persisting as the so-called conus tendon[171,172]. The temporary prominence of the neural crest may explain why its ablation interferes with septation of the outflow tract[166]. Proximally, cardiomyocytes originating in the adjacent ventricular walls are invading the remaining mesenchymal shell of the fused ridges, thus forming a dumbbell-shaped muscular wall between the subaortic and subpulmonary channels (Fig. 7h–n; see refs. [166]). The myocardializing area is continuous on its medial side with the crest of the muscular ventricular septum and on its lateral side with the parietal wall of the myocardial outflow tract[166]. This myocardialization occurs in mammals[139,173] and birds[153]. It progresses from proximal to distal, in other words, opposite to the direction of septation. In mice, myocardialization of the dorsal mesenchymal protrusion and the fused proximal ridges of the outflow tract both seem to be associated with apoptotic activity in the underlying mesenchymal layer[166,174].

Cup-shaped arterial valvar leaflets have formed in the distal part of the muscular outflow tract. The myocardial support provided to the still plump leaflets of the arterial valves may assist their closure. Growth of the myocardium in the middle outflow tract, however, slows further subsequent to the formation of the arterial roots[150]. The position of the arterial valves was previously put at the dog-leg bend[8,154,175]. In reality, the arterial roots, containing the semilunar valvar leaflets, have significant length, and derive from the entire middle portion of the outflow tract. Their proximal boundary corresponds roughly with the dog-leg bend, which is present between the two parts of the myocardial outflow tract until CS16. At CS18, this boundary is marked by the

transition of the thin layer of myocardium that surrounds the middle outflow tract into the much thicker myocardium of the proximal outflow tract. Their distal boundary, the sinutubular junction, corresponds with the distal end of the endocardial ridges (Fig. 7a–g). The coronary arteries originate from an endothelial outgrowth of the aortic base[176,177]. The stem of the left coronary artery is first seen at CS18, whereas that of the right coronary artery forms 2–3 days later at CS19[178–182]. These coronary stems contact a periaortic vascular plexus which, in turn, contacts the ventricular coronary plexus[177]. The main coronary trunks can be located in the relatively thick epicardial areas in the atrioventricular and interventricular grooves.

Distally, the arteries within the pharyngeal arches, along with the dorsal aortas, become increasingly asymmetric in distribution, with regression mostly seen on the right side. The artery of the right 6th arch has disappeared between the origin of the right pulmonary artery and the dorsal aorta. The right pulmonary artery, therefore, appears to arise directly from the pulmonary trunk. The right dorsal aorta itself tapers off between the origin of the 7th segmental artery and the confluence of both dorsal aortas, subsequently disappearing at CS20. Meanwhile, the walls of the large arteries become progressively better organized, which reflects the increased expression of extracellular matrix proteins in this period[183].

**Carnegie stage 20**. For stages 19 through 23, Streeter changed his staging system from one based on qualitative morphological criteria to one based on a more quantitative assessment of organ development[184]. With fewer features changed qualitatively, we have reconstructed only two stages. Stage 20 is reached when ~49 days have passed since conception[13]. The reconstructed specimen is shown in Supplemental Fig. 11. Compared to CS18, only limited differences in the venous part of the heart are seen. The caudal part of the left cardinal, or hemiazygos, vein has disappeared[76]. The atrial appendages have become prominent cranioventral extensions, with the appendages still being similar in size. The stem of the pulmonary veins has further increased in diameter (Supplemental Fig. 16b), with its first division now having acquired a myocardial wall. The myocardial atrioventricular canal still resembles a very shallow funnel, in which superior and inferior atrioventricular cushions are no longer separately distinguishable[185]. Hence, they are depicted as hatched in the reconstruction. The left ventricle has now acquired a thick compact myocardial wall and, concomitantly, a pronounced ventrally pointing apex, while the thinner-walled right ventricle has mainly enlarged radially, extending more forward or cranially than the left ventricle (Fig. 3).

The middle part of the tripartite interventricular foramen, which was located between the crest of the muscular ventricular septum, the rightward margins of the atrioventricular cushions, and the myocardializing ventricular end of the parietal outflow ridge[131], has now closed (Fig. 4). The closure is brought about by the extension of the right-sided margins of the endocardial cushions towards the muscularizing proximal outflow ridges[8,163,186]. The newly formed septum does not myocardialize. It is known, therefore, as the membranous ventricular septum[8,163,187]. The position of the GlN ring identifies the borders of the two persisting parts of the initial interventricular foramen (Fig. 5). Of these, the right atrioventricular junction now occupies a near-frontal plane between the ventricular septal crest medially and the atrioventricular junction laterally, while the left ventricular outlet and the subaortic channel follow a more oblique course between the crest of the ventricular septum and the inner curvature of the heart. Myocardialization of the fused proximal ridges has progressed further distally toward the arterial roots (Figs. 7h–n and 10). The valvar leaflets have become longer

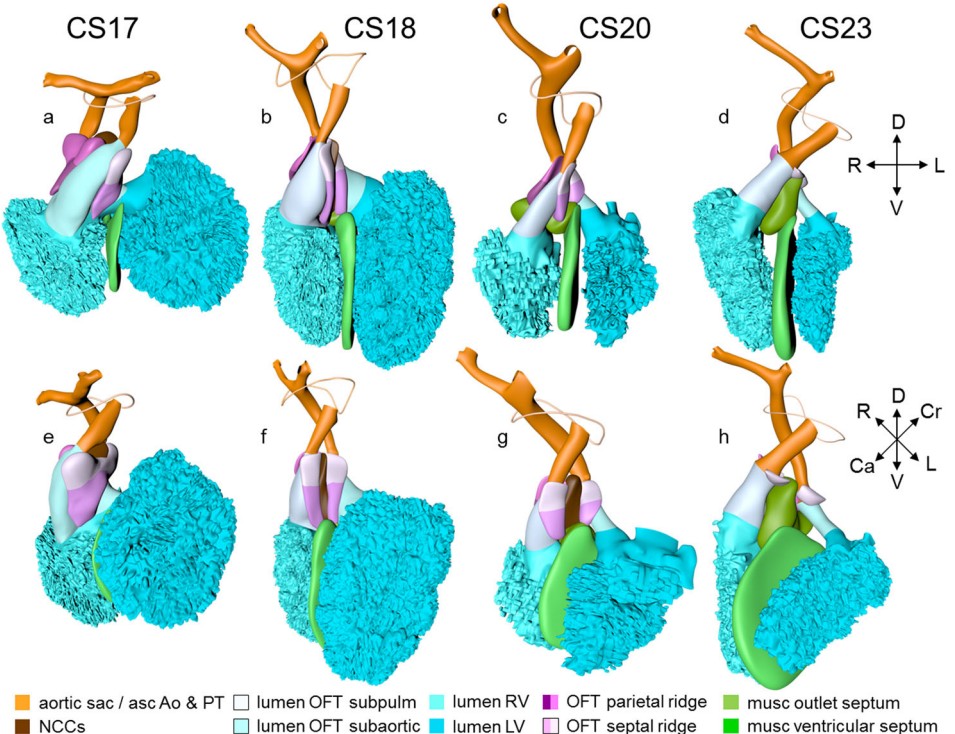

**Fig. 10 Pictorial timeline of the formation of the subpulmonary infundibulum.** Panels **a–d** show cranial views of the lumens of the left and right ventricles, the muscular ventricular septum, the lumen of the outflow tract with ridges and neural crest, and the arterial trunks, while panels **e–h** show left lateral views of the same structures. On completion of the fusion of the proximal outflow tract ridges at CS18, the neural crest cells largely disappear and myocardialization begins (Fig. 7h–n). Myocardialization proceeds from proximal to distal, opposite to the direction of septation, reaching the arterial valves at CS23. The myocardial septum, which then separates the subpulmonary and subaortic channels, also known as the embryonic outlet septum, is located on the right side of the muscular ventricular septum and is topographically part of the right ventricle. This right-sided position accounts for subsequent development into the free-standing muscular infundibulum. At CS23, the base of the subaortic channels is on the left side of the muscular ventricular septum, but the aortic root has not yet been incorporated in to the base of the left ventricle. All images are also available as preset views in the corresponding 3D-PDFs.

and thinner, with the aortic and pulmonary parts separated by the remaining distal part of the neural crest whorl. The arterial walls of the valvar sinuses are beginning to form. The left and right semilunar leaflets of both arterial valves remain within the persisting collar of the outflow tract myocardium. Distal to the developing sinutubular junction, the walls of the ascending aorta and pulmonary trunk have an arterial phenotype. The right-sided dorsal aorta, which was still identifiable as a rudimentary vessel at CS19, has disappeared at CS20.

**Carnegie stage 23**. At this last Carnegie stage, ~56 days have elapsed since fertilization[13]. The reconstructed specimen is shown in Supplemental Fig. 12. Between CS14 and CS23 the diameter of the left sinus horn does not change (Supplemental Fig. 16), implying that it receives an increasingly smaller percentage of the systemic venous blood. Accordingly, the left cardinal vein starts to attenuate between the junction of the left subclavian and jugular veins cranially, and its passage in the left atrial ridge between the left inferior pulmonary vein and the left atrial appendage caudally[188]. Meanwhile, the brachiocephalic vein is forming from merging venous spaces that arise between both jugular veins just cranial to the aortic arch. Remodeling occurs between CS20 and CS21, with only a minute left common cardinal vein present at CS22, the lumen of which has disappeared at CS23[11]. The distal obliterated part of the left sinus horn is known as the ligament of Marshall[189], whereas the remaining proximal part is known as the coronary sinus. The atrial appendages have

increased further in size, and their pectinate muscles are well-developed. Myocardium now surrounds the pulmonary veins up to their second division. The diameter of the stem of the pulmonary veins continues to increase (Supplemental Fig. 16b), preluding the incorporation of the pulmonary veins as 2 separate tributaries into the roof of the left atrium in the 9th week of development and as four tributaries in the 14th week[63]. A fold now begins to form in the roof of the right atrium just rightward of the primary atrial septum (Fig. 4; see ref. [137]). It is against this fold, which is incorrectly known as the secondary atrial "septum", that the primary septum will eventually rest to close the oval foramen.

By now, the leaflets of the atrioventricular valves are forming, although tendinous cords have yet to develop. The leaflets, furthermore, still contain myocardium on their ventricular surface[190]. Fragmentation of the myocardial floor of the tricuspid gully gives atrial blood access to the right ventricular cavity via conduits that pass the septomarginal trabeculation cranially (pre-existing) and caudally (newly formed)[139]. In both ventricles, the papillary muscles begin to form by consolidation of aggregating trabeculations, with the compaction starting at the valve leaflets and moving in the direction of the compact ventricular walls. Epicardially derived cells have begun to induce insulation within the atrioventricular junctions, and have populated the lateral cushions[97,191]. During CS21 and CS22, the whorl of neural crest cells in the ridges of the proximal outflow tract all but disappears, while myocardialization continues. As already explained, the so-called tendon of the conus, a cord-like band between the aortic

and pulmonary roots[171,172], is an inconsistent distal remnant of the whorl. The still long subaortic outlet now passes between the developing mitral valve, the muscular ventricular septum, and the muscularized septum in the proximal outflow tract. The subpulmonary outlet passes between the muscularized septum in the proximal outflow tract and the free right ventricular wall.

The muscular septum in the proximal outflow tract, also known as outlet septum, is normally a temporary embryological structure. It changes in shape and orientation from a dumbbell-like structure perpendicular to the muscular ventricular septum at CS20 to a flat blade almost parallel to the muscular ventricular septum at CS23 (Fig. 10). Extension of its myocardialization towards the developing arterial roots underlies this change in orientation. This positional change coincides with the incorporation of the proximal outflow tract into the ventricles[152,153,156], and transforms the transitory outlet septum into the smooth medial wall of the right ventricle, the dorsocranial part of which becomes the "free-standing" muscular subpulmonary infundibulum[171]. The attribution of the muscular septum of the outflow tract as a mostly right ventricular structure[131] can be best appreciated if the ventricular cavities, muscular septum, and (sub-)aortic and (sub-)pulmonary channels are observed from the left (Fig. 10e–h). The developmental events underlying the transformation of the embryonic outlet septum from a septal to a mural structure are still poorly understood, but probably reflect the asymmetric growth of the increasingly wedge-shaped and transversely oriented right ventricle (Fig. 3). Should the middle portion of the interventricular foramen fail to close, then the result will be a perimembranous ventricular septal defect. Should the asymmetric growth of the right ventricle and the transfer of the aortic root to the left ventricle be hampered, however, the result will be tetralogy of Fallot, or double-outlet right ventricle. In all these settings, it remains possible to recognize a muscular or fibrous outlet septum[192].

Between CS18 and CS23, the arterial valvar leaflets become slenderer, the walls of the sinuses better formed, and the ventriculoarterial and sinutubular junctions identifiable structures. As development progresses, the myocardial cells of the valvar cuff covering the left and right leaflets do not proliferate but become diluted in the surrounding proliferating epicardial connective tissues. Remnants of this myocardium can, nevertheless, persist at least until the 3rd trimester and perhaps into adult life.

**Coda**. We have described the morphological development of the human heart between its first appearance at CS9 up to CS23, when almost all structures of the definitive heart have formed, although at this stage several have still to reach their relative sizes and definitive positions. Because we used embryos that had been carefully staged at the Carnegie Institution without exclusive attention to heart development, we were able to assign critical events in heart development to specific stages of human embryonic development.

Supplemental Table 1 summarizes the relatively recent finding that early heart development progresses by the addition of cardiomyocytes at the venous and arterial periphery of the heart tube (reviewed in ref. [45]). During CS9, the first heart field produces the cardiomyocytes of the embryonic left ventricle and a fraction of those in the atrioventricular canal, atriums, and right ventricle. Ongoing addition of cardiomyocytes from the second heart field to the venous and arterial poles of the first heart field, and differentiation into cardiac compartments, is therefore necessary to form the definitive heart. During CS10, the embryonic right ventricle arises by the addition of cardiomyocytes from the cranial second heart field, followed by the proximal myocardial portion of

the outflow tract during CS11 and the distal myocardial portion during CS12. Meanwhile, the caudal second heart field produces the cardiomyocytes of both atriums during late CS10 and early CS11, while those of the venous sinus are added during CS12. The pharyngeal arch arteries are successively added between CS12 and CS14, while the neural crest cells invade the endocardial ridges of the outflow tract during CS14. Endocardial cushions and ridges form during CS14 to allow for separate blood flows. The non-myocardial distal portion of the outflow tract, from which the arterial trunks form, begins to appear at CS15[121,135].

This "peripheral growth" model of heart development ends when the building plan of the heart has been established. We consider the early ventricles "embryological", because they still stand to receive substantial additional cell contributions. The free wall of the left ventricle receives an extensive contribution of cardiomyocytes from the adjacent part of the atrioventricular canal starting at CS13[87]. The non-trabeculated distal infundibulum ("conal") part of the right ventricle, on the other hand, originates from the incorporation of the smooth-walled proximal outflow tract starting at CS17[152,193]. The central structures in the heart, such as the atrioventricular canal, inner heart curvature, and muscular outflow tract, which temporarily retain their relatively undifferentiated status as remnants of the primary heart tube, subsequently contribute to the internal remodeling that is necessary to achieve septation[112]. The process of septation achieves separate pulmonary and systemic circulations, and is completed by the closure of the middle portion of the interventricular foramen at CS20. It is not known, to our knowledge, whether the completion of septation is associated with a new functional capacity. Since the timing corresponds with the transition of the yolk sac to the (hemo-)chorial placenta[107,194], we submit that the enhanced pumping efficiency or capacity is a determining factor.

The description of the respective developmental stages required an unexpected difference in the number of words needed to delineate stage-specific differences. On average, 500–800 words sufficed to describe the incremental changes in heart development for most stages, but the description of CS12, CS14 and CS(17 + 18) required double that number. The text of CS12 exceeds the average length of a section because it includes the description of the looping of the heart tube. Human CS13 or "Horizon XIII"[184] embryos have developed 30–36 somites[195], which makes them comparable to Theiler stage 16 mouse embryos[196]. This stage is considered "phylotypic" because the basic body plan of vertebrate embryos has been established at this stage and gene-expression profiles between model species of vertebrate groups are most similar[14]. Subsequent to this very conserved stage, CS14 is characterized by the appearance of an array of new features, such as the venous valves, the primary atrial septum, a patent pulmonary vein, the muscular ventricular septum, the ridges in the outflow tract, the cardiac neural crest, the non-myocardial walls of the distal outflow tract, the beginning transformation of the aortic sac into the arterial trunks, and the appearance of the last and special pair of pharyngeal arch arteries[113]. The transition of CS13 to CS14 does not, of course, proceed abruptly, but should be considered as a change in developmental pace.

Marsupial embryos at CS18 have advanced sufficiently to survive outside the womb even though, for instance, their interventricular foramen has yet to close[197]. Perhaps to prepare for extrauterine survival marsupials or for perfusion of the (hemo-)chorial placenta in eutherian mammals, the heart extensively remodels during CS17 and CS18. It is at these stages that fusion occurs between the superior and inferior atrioventricular cushions, and between the septal and parietal ridges of the outflow tract. During the same period, the interventricular foramen remodels to accommodate unimpeded systemic and pulmonary blood flows, the mesenchymal components of the

atrial septum and the outlet septum myocardialize to buttress the structures to which they contribute, the coronary arteries form to nourish the newly formed compact wall of the ventricles, the muscular outflow tract remodels with the incorporation of its proximal part into the ventricles, and a start is made with the development of valves. It is closure of the middle part of the interventricular foramen by the formation of the membranous septum at CS20 that completes septation.

Based on the features described in the previous paragraphs, we hypothesize that embryonic heart development includes an early phase, during which its basic building plan is laid down, in accordance with the peripheral growth model. There is then a later phase during which the heart remodels to cope with the requirements of postnatal or placental circulation in marsupial and eutherian mammals, respectively[107,194]. The subsequent fetal phase varies markedly in length between species. It can, in the case of marsupials, even be nonexistent, with development proceeding in a pouch, which is outside the womb.

The mouse embryo is the most frequently used animal model for (experimental) studies of heart development. Accordingly, mouse data is also the main reference for mechanistic extrapolation in this study. Almost invariably, murine development is not based on a staging system, such as that of Theiler[196]. Instead, it is expressed in embryological days ("E" or "ED"), which refers to the time since midnight before demonstrating the presence of a vaginal plug in the morning. The method is prone to variation, as evidenced by the frequently reported inter- or even intra-litter variation of half a day or more[198,199], which is the equivalent of one Carnegie stage (Supplemental Fig. 13). The relation between gestational age and developmental stage is even weaker in human embryos[200]. Such errors are the more pressing as early heart development in mice is considered to progress rapidly relative to human development. To address these issues several, mutually slightly differing staging systems have been developed, which focus on gastrulation or early heart development[19,22,52,201]. However, even differences in the rate of development between structures within a single embryo occur. Well-known examples are that between somite and heart development in mice[202], and that between different structures in the developing heart of human embryos[11]. The relation between the rate of development of human and mouse embryos in the time frame that we describe is, therefore, an important issue. To determine the developmental effects of radiation toxicity, Otis and Brent used the first appearance of 137 structures, most of which in the nervous, digestive, or circulatory systems, to establish "equivalent ages" in mouse and human embryos[198]. For our study, the period between ED8 and ED16.5 is most interesting. That period encompasses a phase (ED8-13.5, equivalent to CS9-20) in which the daily development of the mouse embryo is approximately fourfold faster than that of the human embryo, and a second phase (ED13.5-ED16.5) in which daily development of the mouse embryo exceeds that of the human embryo ~13-fold[198]. Unfortunately, early heart development is at the very beginning of this comparison. To validate Otis and Brent's comparison for that early period, we staged our reconstructed early human hearts according to Le Garrec's system for early mouse hearts[52]. Supplemental Fig. 17 shows that our data confirm those of Otis and Brent[198], and that early heart development in mice and humans proceeds at the same rate as subsequent development. Apparently, rapid overall development and insufficiently careful staging of mouse embryos are to blame for the perception that early heart development in the mouse differs from that in humans.

**Quantitative morphology**. We have taken great care to calibrate the scale cubes that we added to each of the reconstructions. The scale cubes, therefore, permit comparisons of structures between stages in real size. They can be used to settle arguments of the effects of differential growth on shape. Especially in cardiac structures with components that differ greatly in growth rate, such as the respective parts of the outflow tract, proportional comparisons are useful. The pictorial timelines revealed, for instance, that the absence of growth in the myocardial outflow tract was compensated for by growth of the intrapericardial arterial trunks, in particular the aortic trunk (Fig. 6c–d). These comparisons further showed that the formation of the semilunar leaflets of the arterial valves started with a selective increase in the diameter of the distal endocardial ridges at CS16, the forming of a dividing furrow at CS17, and the division into aortic and pulmonary roots at CS18. Such assessments, therefore, allow a coherent account to be advanced regarding the development of the arterial pole of the heart.

Attention to the segmental structures, such as the somites and spinal ganglia, and longitudinal structures, like the dorsal aorta, gut, and central nervous system, in the reconstructions were instrumental in determining changes in the relative positions of organ structures. In particular developmental changes in the helical course of the wall of the heart tube, parts of which have been controversial for a long time, could be measured accurately. During the looping phase of heart development (CS10-CS12), the helical course of the wall was clockwise in direction when following the bloodstream (Fig. 2). Following this pre-pattern, the endocardial ridges of the outflow tract, which made their appearance during CS14, spiraled clockwise. During the next 3 stages (~1 week), this spiraling course was reversed[141,142], becoming transferred to the intrapericardial arterial trunks by the obliquely growing aortopulmonary septum (Fig. 9a), in other words to non-myocardial structures. This unwinding precedes the extensive remodeling of the myocardial outflow tract between CS16 and CS18, but whether there is a relation between these structures remains to be established.

**Limitations of the study**. We have provided 12 detailed reconstructions of human embryonic hearts between CS9 and CS23. Although it can reasonably be stated that 12 models cannot visualize all of the cardiac development, we were able to provide a continuous account of the changes in size and shape of the heart. We did not encounter major gaps between the models in the sense that the transition of younger configurations into older arrangements could be described without having to assume major steps in development that are not supported by data. A valid question, nevertheless, is whether we have accounted for all variation. Although the answer is obviously "no", differences between specimens could usually be explained as small differences in the degree of individual development rather than a deviation from the expected morphology. An example is the CS9 model in our series. This embryo (Carnegie #3709) has four or five somites, which places it in the least advanced CS10 group, but its cardiovascular development is least developed in the entire series of early hearts described by Davis[15]. Questions can, therefore, be posed with regard to the normality of this embryo. Since two additional embryos with five formed somites show a near-identical morphology with respect to its cardiovascular system[9,33], we assume that the reconstruction represents a normal stage of heart development. Our findings in human embryos fall in line with earlier observations in mice, revealing that heart and early somite development do not proceed strictly synchronously[52,202]. Another example is the appearance of the pulmonary arteries in our model embryo at CS15, whereas Sizarov et al. associated their appearance with CS14[48]. In the Carnegie collection, the pulmonary arteries make their appearance in ~75% of the 44 embryos at CS14 and in

the remainder at CS15[11]. Such data indicate that small inter-individual differences exist in the developmental timing of organogenesis. The most important limitation of the present series is probably that the models still contain mistakes. Because the models are made in the software program Cinema 4D, such mistakes can be corrected relatively easily. We, therefore, encourage readers to report such errors.

## Methods

**Embryos**. This study was undertaken in accordance with the Dutch regulations for the proper use of human tissue for medical research purposes. Staged human embryos were obtained from the Digitally Reproduced Embryonic Morphology (DREM) project (Dr John Cork; Cell Biology & Anatomy, LSU Health Sciences Center, New Orleans; https://www.ehd.org/virtual-human-embryo/about.php, http://virtualhumanembryo.lsuhsc.edu/DREM/OnlineDisks.html). These embryos are part of the Carnegie collection, Washington D.C., USA. We reconstructed 12 hearts from human embryos obtained between ~26 and ~56 days of development subsequent to fertilization (Supplemental Table 2). In addition to the reconstructed and modeled embryos, we also studied the immunohistochemically stained sections of human embryonic hearts collected and produced by Viragh and Wessels[94,203,204], Sizarov[33,48,128,136,205], and Ya[33]. Finally, we inspected the digitized embryos of the Blechschmidt collection[206] to study the prevalence of venous congestion in the embryos of that collection. We used the criteria of O'Rahilly, as modified in 2010[13], to correlate Carnegie stages of human development with days of development subsequent to fertilization. The description of developmental processes in human embryos is, where appropriate, underscored with experimental data from other mammals, in particular mice, and if fitting, also with data from chickens. Theiler's staging system[196] was used to correlate the stages in murine development with the Carnegie stages. Theiler used Streeter's Horizons[184,207] to stage his mouse embryos. Streeter's Horizons XI-XXIII were absorbed virtually unchanged into O'Rahilly's Carnegie stages 11–23[19,81,87], but Carnegie stages 8, 9, and 10 correspond with Horizons VIII-IX, X/a, and X/b, respectively[81,207]. Appendix I of Kirby's monograph on cardiac development[208] was used to correlate Hamilton and Hamburger's staging system of chicken embryos[209] to the Carnegie stages (Supplemental Fig. 13). Heart development in chicken embryos was carefully tabulated by Martinsen[210], but this study does not systematically correlate chicken to mammalian development. Segmental levels in the embryo were determined perpendicular to the curvature of the spinal cord. Segmental levels were related to somite number up to CS13, and to spinal ganglion number from CS14 onwards. Because the occipital somites do not induce spinal ganglia, the latter number is 4 units smaller.

**Image acquisition, 3D reconstruction, and visualization**. Processing of the digital images, and calculation of voxel size, were performed as described previously[76]. AMIRA (version 2020.2; FEI Visualization Sciences Group Europe, Merignac Cedex, France) was used to generate 3D reconstructions. Preliminary alignment of consecutive sections was performed automatically with the least-squares method, followed by further manual alignment. The definitive alignment also accounted for curvatures in the sagittal and transverse planes of the body axis of the embryo. Existing images of the embryo before sectioning or age-matched embryos that were imaged with magnetic resonance were used as the template for proper alignment (e.g., http://embryo.soad.umich.edu, http://virtualhumanembryo.lsuhsc.edu/DREM/OnlineDisks.html). Delineation of heart structures was performed manually, using the immunohistochemically stained sections described elsewhere[33,48,94,128,136,203–205] as guides.

Polygon meshes from all reconstructed materials were exported via "vrml export" to the remodeling software Cinema 4D (version R21; MAXON Computer GmbH, Friedrichsdorf, Germany). The accuracy of the remodeling process was safeguarded by simultaneous visualization in Cinema 4D of the original output from Amira and the remodeled Cinema model. Subsequently, the Cinema 3D-model was exported via "wrl export" to Adobe's portable device format (PDF) reader version 9 (http://www.adobe.com) for the generation of 3D-interactive PDF files (Supplemental Figs. 1–12, an instruction video is included to show how such interactive 3D-PDF works). The list of structures identified in each of the reconstructed embryos and their color code are summarized in Supplemental Table 3. An overview of the workflow from the raw 3D reconstruction in Amira via the polished 3D configuration in Cinema 4D to the interactive 3D-PDF is shown in Supplemental Fig. 19.

**Measurements**. For all goniometric measurements, the structures of interest were aligned using long craniocaudal structures, such as the dorsal aorta or neural tube (Supplemental Fig. 18). All axial lengths were measured in the Cinema reconstructions with the "spline" function, which takes the curvature of structures into account. Dots in the graphs represent measured values of a single embryo, while the lines connecting the observed values were constructed manually.

**Description**. The descriptions of the embryonic hearts follow the sequential segmental approach[211]. We use cranial-caudal, dorsal-ventral, and left-right to describe topographical relations, with the cervical and upper thoracic vertebral column being dorsal. The terms proximal and distal refer to positions relative to the center of the heart.

**Reporting summary**. Further information on research design is available in the Nature Research Reporting Summary linked to this article.

## Data availability

All data generated or analyzed during this study are included in this published article (and its supplementary information files). Supplemental Tables and Figures are deposited in Figshare, which is a recognized public repository. The hyperlink of each Supplemental Table/Figure is made available in the main text, the Supplementary PDF and summarized as follows. Supplemental Table 1: https://doi.org/10.6084/m9.figshare.19102571; Supplemental Table 2: https://doi.org/10.6084/m9.figshare.17144156; Supplemental Table 3: https://doi.org/10.6084/m9.figshare.17121977; Instruction video Supplemental Figs. 1–12: https://doi.org/10.6084/m9.figshare.17033225; Supplemental Fig. 1: https://doi.org/10.6084/m9.figshare.17033156; Supplemental Fig. 2: https://doi.org/10.6084/m9.figshare.17033189; Supplemental Fig. 3: https://doi.org/10.6084/m9.figshare.17033249; Supplemental Fig. 4: https://doi.org/10.6084/m9.figshare.17033252; Supplemental Fig. 5: https://doi.org/10.6084/m9.figshare.17033255; Supplemental Fig. 6: https://doi.org/10.6084/m9.figshare.17085503; Supplemental Fig. 7: https://doi.org/10.6084/m9.figshare.17085506; Supplemental Fig. 8: https://doi.org/10.6084/m9.figshare.17085509; Supplemental Fig. 9: https://doi.org/10.6084/m9.figshare.17085512; Supplemental Fig. 10: https://doi.org/10.6084/m9.figshare.17085515; Supplemental Fig. 11: https://doi.org/10.6084/m9.figshare.17085521; Supplemental Fig. 12: https://doi.org/10.6084/m9.figshare.17085524; Supplemental Fig. 13: https://doi.org/10.6084/m9.figshare.17143910; Supplemental Fig. 14: https://doi.org/10.6084/m9.figshare.19102616; Supplemental Fig. 15: https://doi.org/10.6084/m9.figshare.17143970; Supplemental Fig. 16: https://doi.org/10.6084/m9.figshare.19102622; Supplemental Fig. 17: https://doi.org/10.6084/m9.figshare.17144036; Supplemental Fig. 18: https://doi.org/10.6084/m9.figshare.17144090; Supplemental Fig. 19: https://doi.org/10.6084/m9.figshare.17144114.

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

## Acknowledgements
We thank Dr. John Cork (New Orleans) for making extra sections of embryos included in the DREM collection available and Dr. Antoon Moorman (Amsterdam) for constructive discussions and comments. Furthermore, we would like to thank our bachelor students, who worked on heart development as part of their academic internship. The financial support of "Stichting Rijp" is gratefully acknowledged.

## Author contributions
J.P.J.M.H. was responsible for data collection, analysis, and visualization. N.K., S.E.K., and W.H.L. participated in data analysis and interpretation. J.P.J.M.H., N.K., and G.M.C.M. were responsible for the reconstruction and modeling of the 3D-PDFs. S.E.K and R.H.A. participated in data analysis and interpretation, provided guidance, and edited the manuscript. J.P.J.M.H. and W.H.L. conceived the study and wrote the manuscript.

## Competing interests
The authors declare no competing interests.
