## [Peer Review File · Communications Biology]

Reviewers' comments:

Reviewer #1 (Remarks to the Author):

The manuscript by Hikspoors et al., offers a pictorial timeline of morphogenesis of human cardiovascular system from CS9-CS23. The comprehensive morphometric analyses highlight age-dependent events, while graphs visualize growth and spiraling of the wall of the heart tube. The authors had several interesting observations including the relative growth of the pulmonary trunk and the aortic trunk of the outflow tract. Overall, this is a major effort to resynthesize the literature and to display the complex structures in Cinema4D models. A major concern of the manuscript is many of the findings have been reported previously. Nevertheless, it will be of interest to others in the field of cardiovascular development and the wider field of developmental biology and human birth defects.

I have several minor concerns regarding the manuscript:

1. In Figure 2 and 6, similar colors were used to represent different structures. There are very difficult to tell apart, especially on the print.
2. In Figure 7, the graphs display changes of distance. What about volume? For example, could the apparent decrease of the myocardial outflow tract length is in fact coupled with an increase of width and/or volume?
3. What is the reference for the non-myocardial mural cells in Fig.8 and 9?
4. The observation of the aortic trunk continues to grow while the pulmonary trunk does not is very intriguing. Why does pulmonary trunk fail to grow?

Reviewer #2 (Remarks to the Author):

The work presented by Hikspoors and authors provides a novel and exquisitely detailed construction of human heart development starting from a remarkably early point during embryonic development. Currently, there is a great need in the field of cardiac developmental biology to translate findings from animal models such as mouse and chick to understandings of human heart development. While extensive work has been done using murine embryonic models, translation of morphologic events to human heart development has been largely speculative. By taking human samples and constructing an extensively annotated 3D model across multiple timepoints, the authors have provided a rare opportunity to visualize human heart development from an early timepoint. Most surprisingly, the author's reconstruction of gastrulation stage embryos at CS9 provided an intriguing glimpse into the structures present in pre-heart tube human embryos and provided a description that is uniquely similar to the structures found in murine and chick models. Importantly, the authors contextualized each stage of development with quantification of somite number, and key developmental structures which facilitates correlation to other mammalian models of development and cross referencing across human embryos. The inclusion of vascular development in the 3D reconstruction also provides a contextual understanding of the development of the pulmonary and systemic circulatory systems which created an excellent and clear visual representation of cardiovascular development throughout the embryo.

Importantly, the authors provided intriguing insights into the spatiotemporal development of individual cardiac chambers. For example, the authors comment that the left ventricle develops trabeculations earlier than the right ventricle and appears to have differing morphologic features due to temporal separation in chamber development. Moreover, the observations of asymmetric proliferation is consistent with previous studies in murine models of heart looping and ballooning of the left and right ventricles.

Overall, the authors have provided a novel and exquisitely detailed 3D reconstruction of human heart development that will provide an extraordinary tool for cardiac developmental biologists and cardiac anatomists. While the focus of this paper has been to describe human cardiac development, a greater discussion on the relations to murine models would heighten the impact of the paper by providing a reference point for understanding similarities and differences between both model systems.

Specific Comments:

1. The authors describe *Mesp1* as a marker of the first heart fields in lines 74-77. In situ hybridization studies and scRNA-seq studies in mouse (see Lescroart et al, Science, 2018 <https://science.sciencemag.org/content/359/6380/1177.editor-summary> and Pijuan-Sala et al 2019 <https://www.nature.com/articles/s41586-019-0933-9>) have shown this to be a mesodermal progenitor marker present during early gastrulation. After migration to the anterior lateral plate mesoderm, *Mesp1* is downregulated as cardiac progenitors are specified into either first or second heart fields. *HCN4* and *TBX5* are more appropriate markers to describe the FHF.
2. The authors should clarify whether "cardiac primordium" (lines 83-84) refers to the heart fields such as the cardiac crescent. Based on supplementary figure 2, the crescent like structure resembles the early cardiomyogenic FHF present at embryonic day 7.5 in the mouse.
3. Color labeling schemes used proved to be confusing for the reader due to the similarity in color between closely positioned structures. For example, in Figure 3, the lumen of the LV and RV are closely related in a teal color as well as the OFT lumen. This produced confusion in terms of understanding the border at which each structure starts and end.
4. Authors provide a discussion of endocardial cushion development in figure 5. While figure 5 provides a visual representation of the closure of the atrial foramen and ventricular septation, it is unclear how the endocardial cushions ultimately contribute to the closure of the atrial foramen. Color scheme creates difficulty in separating distinct structures present. Due to the blue colors between the lumen of the LA, LV, and OFT, it is unclear how the cushions ultimately close the atrial foramen given the septation of the outflow tract as they appear to be adjacent but not directly involved in closing the foramens.
5. Figure 9, the authors describe the role of the neural crest in the separation of the pulmonary and aortic trunks. The bottom panel shows a progressive separation of the lumens pulmonary and systemic arterial trunks. The separation of the lumens into two distinct trunks is unclear from the image shown. While the neural crest appears to run parallel and spiral with the trunk, the image does not demonstrate the invasion that ultimately create the separation between pulmonary and systemic outflows seen by CS18. Greater clarity in how a seemingly unified lumen during early development is separated into two distinct vessels is needed in the figures.
6. While the 3D PDFs allow for user friendly exploration of the distinct structures present during development, the authors may consider creating a time lapse video for describing complex septation processes and valvular development. Particularly those events described in figure 8 and 9, the septation of the outflow tract both in the proximal muscular segments and distal vascular segments are unclear based on the parallel endocardial ridges and neural crest structures that are shown in the figures.

Reviewer #3 (Remarks to the Author):

The authors have reconstructed 12 specimen from the Carnegie collection of histological sections, providing a 3D atlas of human embryonic development, with annotated segmentations of cardiac structures. Outstanding 3D-pdfs are provided, for an interactive visualization. 12 figures recapitulate the temporal progression of specific features. The text describes each anatomical stage and refers to mechanistic insights obtained in the experimental mouse model. Overall, this is a very careful, rather complete and well-referenced study, which will be useful for anatomists, developmental biologists, pediatric cardiologists, as well as students. I have minor comments to clarify aspects of the study.

- 1- How the structures have been segmented based on histological features is not fully clear. Segmentation is an interpretation, and it is important to show examples of how the contours superimpose on raw data. It is unclear why the arterial most part of the early heart is referred to as "outflow tract" in the text and 3D-pdfs of CS10-11, when the authors elsewhere (ex line 970) and in the Table acknowledge that the RV is added before the OFT to the tube. In the mouse, the early heart tube similar to that shown at CS10 was shown to contain LV and RV cells (Zaffran et al, 2004, ref 43).
- 2- I do not understand how Supplementary Figure 1 has been drawn. Which criteria were used for the inter-species comparison ? Please illustrate the comparisons, for example using the HREM images of Mohun and Anderson. I am surprised to see a linear relationship : (1) Development

doesn't have to be proportional between species ; (2) In this graph, human stages are based on morphological features whereas mouse stages are based on time. To me CS9 (shown in Ref 9 or 24) to CS12 all correspond to successive looping stages at E8.5 in the mouse. Others have proposed a looser correlation (Krishnan et al., 2014).

3- The quantitative analyses should be treated with a bit more caution, as based on single individuals. The legend should explain that one dot is one sample. The paragraph. "differences between specimens could usually be explained as small differences in degree of individual development rather than deviation from the expected morphology" is not a conclusion that can be reached before quantitative parameters are measured on a batch of embryos. CS9 shown here is for example very different from CS9 shown in Ref 9 or 24.

4- " We did not encounter major gaps in our description of the models ". Without diminishing the quality of their study, the authors should acknowledge that early heart development is barely covered, from the cardiac crescent to tube formation and to tube looping. These stages have been shown in the mouse to be very dynamic (Tyser et al 2006, Tyser et al 2021, and ref 43) and would require segmentation of more CS9-CS12 embryos to be fully caught.

5- Another limitation of the approach based on histological sections, is that the z-resolution is poor, as well as the registration of sections.

6- It is a pity that CS15 could not be reconstructed entirely. Could an alternative sample be found, like the one segmented in Ref 9 or from another embryo collection ?

7- Anatomical descriptions are very useful. However, a caveat is to draw mechanistic conclusions which can instead only result from experimental conditions. See for example lines 170 or 570-571, which are confusing in this regard. In the text, the authors should make it clearer when they refer to previous mouse experimental data, compared to their own original anatomical observations.

Example : "The first heart field, which is defined during CS8 ref 16" is incorrect. Do the authors mean "The first heart field, which has been shown in the mouse to arise at a stage equivalent to CS8 " ? See also line 195

8- The statement "that the first heart field (..) gives rise primarily to the embryonic left ventricle" is wrong. Clonal data (ref 63) clearly show that it also contributes to the RV, the AVC and the atria. Ref 19, 20 are based on genetic tracing, which is not an appropriate method to trace the lineage of cells (Ref 36).

9- The statement that the heart starts beating at CS10 is surprising. In the mouse, cardiomyocytes beat as soon as they differentiate, and so already at the crescent stage (Tyser et al., 2006)

10- Since the text refers a lot to mouse data, it would be useful to have a paragraph discussing differences in the two species. I was surprised by (1) the massive jelly at CS9, (2) the lack of outflow cushions/ridges at CS13, whereas AVC cushions are segmented, (3) the extension of AVC cushions into the atrium. Are these human features ?

11- "The terms proximal and distal refer to positions relative to the center of the heart" : isn't it rather referring to the sense of blood circulation ?

12- Please explain in the legend of the CS9 3D-pdf the green stripes. Please annotate the aorta as distinct from the pulmonary trunk in Figs 7-10, and the fishmouse in Fig. 7

Typo

Line 124 and 224 : rightward

Line 397 : is GIN2 equivalent to GIN ?

The link <https://www.prenatalorigins.org/virtual-human-embryo> doesn't work

Response to reviewers

COMMSBIO-20-3607A

A pictorial account of heart development: spatial and temporal aspects of the human embryonic heart between 3.5 and 8 weeks of development

Reviewer #1 (Remarks to the Author):

The manuscript by Hikspoors et al., offers a pictorial timeline of morphogenesis of human cardiovascular system from CS9-CS23. The comprehensive morphometric analyses highlight age-dependent events, while graphs visualize growth and spiraling of the wall of the heart tube. The authors had several interesting observations including the relative growth of the pulmonary trunk and the aortic trunk of the outflow tract. Overall, this is a major effort to resynthesize the literature and to display the complex structures in Cinema4D models. A major concern of the manuscript is many of the findings have been reported previously. Nevertheless, it will be of interest to others in the field of cardiovascular development and the wider field of developmental biology and human birth defects.

I have several minor concerns regarding the manuscript:

1. In Figure 2 and 6, similar colors were used to represent different structures. There are very difficult to tell apart, especially on the print.

This question was also raised by reviewer 2. We have used >70 color codes. We further avoided the use of dark colors, because they diminish the perception of depth considerably, especially when printed. The combination makes it unavoidable that some colors are close. We have opted to show similar structures, such as the lumen or myocardial coat of the heart, in nuances of the same color. We would like to adhere to that approach. We have tried, nonetheless, to increase the difference between these nuances. We hope the reviewer can agree with our modified palette.

2. In Figure 7, the graphs display changes of distance. What about volume? For example, could the apparent decrease of the myocardial outflow tract length in fact be coupled with an increase of width and/or volume?

It was already established 25 years ago that the myocardium of the outflow tract grows less rapidly than the ventricular myocardium (Knaapen et al., 1995). Unfortunately, these authors subdivided the heart in a way that does not fit well with that which we present here. After our study was submitted, Faber and Jensen (2021) have published quantitative data on segments of the developing human heart that are comparable to our segments. They report that the OFT stops growing in volume at ~32 days of development). Please note that Faber and Jensen have related Carnegie stages to post-fertilization ages of the embryos according to O'Rahilly, (1971a), whereas we used O'Rahilly's modified 2010 correlation (O'Rahilly and Muller,

2010). Based on O'Rahilly (1971a), 32 days corresponds to CS14. At the request of the reviewer, we have also measured growth of the myocardium of the outflow tract in our reconstructions, and have compared it to that of both ventricles. We find that the growth rate of the myocardium of the outflow tract and ventricles is similar between CS12 and CS15, and that their volumes approximately double in that period. Thereafter, the growth rate of the ventricles slows down ~3-fold, while that of the outflow tract declines >10-fold. This abrupt decline of growth coincides with the 2-3-fold decline in the length of the outflow tract and, therefore, demonstrates that the wall of the outflow tract mainly changes in shape. We have now added this Figure and the accompanying text to the manuscript in the section on CS17.

3. What is the reference for the non-myocardial mural cells in Fig.8 and 9?

We are not completely sure what the reviewer means with “reference”. The shape of the columns of neural crest cells can be deduced from the denser packing of the neural crest cells in the endocardial ridges, as described in some of our earlier work (Webb et al., 2003, Anderson et al., 2003, Anderson et al., 2012), as well as in that of some of our students (Yang et al., 2013) or colleagues (Okamoto et al., 1981, Orts-Llorca et al., 1982, Thompson et al., 1985, Bartelings and Gittenberger-de Groot, 1988). The pronounced changes in the configuration of the outflow tract from a long and slender structure into a much shorter and plumb structure became visible in our reconstructions. This change in shape has attracted our attention for ~15 years, but only after inspecting our interactive reconstructions, we realized what was going on.

4. The observation of the aortic trunk continues to grow while the pulmonary trunk does not is very intriguing. Why does pulmonary trunk fail to grow?

The “why” question is difficult to answer at this moment, but, as we describe in the sections on CS14-17, we found a temporal correlation between the abruptly declining growth of the pulmonary trunk and the disappearance of the well-delineated, Isl-1-positive peritracheal mesenchyme that is continuous with the pulmonary trunk. The studies described in Yang et al. (2013), Liang et al. (2014), Zhou et al. (2017), Jin et al. (2019), and Li et al. (2019) have shown that the wall of the pulmonary trunk is made up predominantly of Isl-1-expressing cells. In contrast, the wall of the ascending aorta is predominantly made up of neural crest cells (Harmon and Nakano, 2013, Sawada et al., 2017, Jin et al., 2019). We have addressed the difference in growth characteristics of the ascending aorta and pulmonary trunk in the section on CS16.

Reviewer #2 (Remarks to the Author):

The work presented by Hikspoors and authors provides a novel and exquisitely detailed construction of human heart development starting from a remarkably early point during embryonic development. Currently, there is a great need in the field of cardiac developmental biology to translate findings from animal models such as mouse and chick to understandings

of human heart development. While extensive work has been done using murine embryonic models, translation of morphologic events to human heart development has been largely speculative. By taking human samples and constructing an extensively annotated 3D model across multiple timepoints, the authors have provided a rare opportunity to visualize human heart development from an early timepoint. Most surprisingly, the author's reconstruction of gastrulation stage embryos at CS9 provided an intriguing glimpse into the structures present in pre-heart tube human embryos and provided a description that is uniquely similar to the structures found in murine and chick models. Importantly, the authors contextualized each stage of development with quantification of somite number, and key developmental structures which facilitates correlation to other mammalian models of development and cross referencing across human embryos. The inclusion of vascular development in the 3D reconstruction also provides a contextual understanding of the development of the pulmonary and systemic circulatory systems which created an excellent and clear visual representation of cardiovascular development throughout the embryo.

Importantly, the authors provided intriguing insights into the spatiotemporal development of individual cardiac chambers. For example, the authors comment that the left ventricle develops trabeculations earlier than the right ventricle and appears to have differing morphologic features due to temporal separation in chamber development. Moreover, the observations of asymmetric proliferation is consistent with previous studies in murine models of heart looping and ballooning of the left and right ventricles.

Overall, the authors have provided a novel and exquisitely detailed 3D reconstruction of human heart development that will provide an extraordinary tool for cardiac developmental biologists and cardiac anatomists. While the focus of this paper has been to describe human cardiac development, a greater discussion on the relations to murine models would heighten the impact of the paper by providing a reference point for understanding similarities and differences between both model systems.

We thank the reviewer for his/her positive evaluation. We have based all our mechanistic considerations on data from rodents and, if they were not or insufficiently available, from chicks. In our experience, the similarities dwarf the differences. Apart from the regression or persistence of the superior caval vein (persistence is at ~0.4% fairly common in humans (Bergman's comprehensive encyclopedia of human anatomical variations, 2016), the main difference between both mammalian species may reside in an accelerated morphogenesis of the very early rodent heart. We have studied this issue and have found that this difference is perceived rather than real. We describe this finding now in our Discussion.

Another difference between rodents and humans is the remodeling of the left superior caval vein into the brachiocephalic vein in human embryos. We have addressed the regression of the left superior cardinal/caval vein and the appearance of the brachiocephalic vein in our description of Carnegie stage 23. Fortuitously, we could follow the remnant of the vein over its entire course. As we report in the section on CS23, there is no difference in the course of this vessel between mice and humans.

Specific Comments:

1. *The authors describe Mesp1 as a marker of the first heart fields in lines 74-77. In situ hybridization studies and scRNA-seq studies in mouse (see Lescroart et al, Science, 2018 <https://science.sciencemag.org/content/359/6380/1177.editor-summary> and Pijuan-Sala et al 2019 <https://www.nature.com/articles/s41586-019-0933-9>) have shown this to be a mesodermal progenitor marker present during early gastrulation. After migration to the anterior lateral plate mesoderm, Mesp1 is downregulated as cardiac progenitors are specified into either first or second heart fields. HCN4 and TBX5 are more appropriate markers to describe the FHF.*

We thank the reviewer for this comment. Although we did not cite Lescroart's 2018 Science article, their abstract reiterates earlier statements of their group that "Mouse heart development arises from Mesp1-expressing cardiovascular progenitors (CPs) that are specified during gastrulation", and concludes that "Mesp1 is required for the exit from the pluripotent state and the induction of the cardiovascular gene expression program". We had interpreted this and similar lines as arguing in favor of a critical role for Mesp1 in the cardiovascular differentiation program. Pijuan-Sala's 2019 article does not mention Mesp1 and focuses, in our opinion, on endodermal development. We assume the reviewer points to the fact that Mesp1-expressing gastrulating cells do not give rise exclusively to the first heart field, but also to endocardial cells and cells of the second heart field (Lescroart et al., 2018, Chan et al., 2013). We agree with that assessment and have replaced Mesp1 by Tbx5 in the section describing CS9. We have additionally modified the surrounding text to some extent.

2. *The authors should clarify whether "cardiac primordium" (lines 83-84) refers to the heart fields such as the cardiac crescent. Based on supplementary figure 2, the crescent like structure resembles the early cardiomyogenic FHF present at embryonic day 7.5 in the mouse.*

We apologize for the ambiguity of our text. We discuss CS9 embryos with 5 somites, so the term "cardiac primordiums" is meant to encompass the entire heart as it is at this stage. In addition to the somites, the reconstruction shows vessels, a neural plate, a transverse septum and a coelom, so the embryo has passed beyond the FHF stage. We estimate the heart to be Le Garrec's stage E8.5d (Le Garrec et al., 2017). At that stage, troponin is accumulating in the cardiac wall (Sizarov et al., 2011), in agreement with mouse data (Ivanovitch et al., 2017). To address the comment of the reviewer, we have changed "cardiac primordiums" into "early heart". The reviewer estimates the heart to be "early cardiomyogenic FHF". Reviewers 1 and 3 have also asked for a closer comparison between early heart development in mice and humans. We address this issue now in our Discussion and Supplemental Figure 15.

3. *Color labeling schemes used proved to be confusing for the reader due to the similarity in color between closely positioned structures. For example, in Figure 3, the lumen of the LV*

and RV are closely related in a teal color as well as the OFT lumen. This produced confusion in terms of understanding the border at which each structure starts and end.

This question was also raised by reviewer 1. We have used >70 color codes. We further avoided the use of dark colors, because they diminish the perception of depth considerably. The combination makes it unavoidable that some colors are close. We have opted to show similar structures, such as the lumen or myocardial coat of the heart, in nuances of the same color. We would like to adhere to that approach. We have tried, nonetheless, to increase the difference between these nuances. We hope the reviewer can agree with our modified palette.

4. Authors provide a discussion of endocardial cushion development in figure 5. While figure 5 provides a visual representation of the closure of the atrial foramen and ventricular septation, it is unclear how the endocardial cushions ultimately contribute to the closure of the atrial foramen. Color scheme creates difficulty in separating distinct structures present. Due to the blue colors between the lumen of the LA, LV, and OFT, it is unclear how the cushions ultimately close the atrial foramen given the septation of the outflow tract as they appear to be adjacent but not directly involved in closing the foramens.

In our descriptions of CS13 and CS14, we have defined the cushions surrounding the primary atrial foramen as extensions of the atrioventricular cushions. Accordingly, the cushions in the atrioventricular canal and their atrial extensions fuse concurrently in CS17. Basically, the primary atrial foramen closes due to the apposition and fusion of the superior and inferior cushions (orange). We have slightly modified our text and have indicated the contour of the atrioventricular canal in the Figure, so that the atrial extensions of the cushions become better recognizable. Finally, we have also increased the “weight” of the contour lines of the foramens.

5. Figure 9, the authors describe the role of the neural crest in the separation of the pulmonary and aortic trunks. The bottom panel shows a progressive separation of the lumens pulmonary and systemic arterial trunks. The separation of the lumens into two distinct trunks is unclear from the image shown. While the neural crest appears to run parallel and spiral with the trunk, the image does not demonstrate the invasion that ultimately creates the separation between pulmonary and systemic outflows seen by CS18. Greater clarity in how a seemingly unified lumen during early development is separated into two distinct vessels is needed in the figures.

The reviewer is correct in his/her comment that the separation of the lumen of the distal outflow tract is poorly visible in the CS17 panel of Figure 9. However, the CS18 panel shows an almost complete separation of the aortic and pulmonary channels. This issue arose because we wanted to show the movement of the pulmonary channel around the aorta, while keeping the position of the aorta fixed. We ask the reviewer to look at the same images seen from the other side in Figure 10. Fig 10, panels CS17 and CS18, shows the separation of both channels well, but here, both incipient channels are difficult to see separately in CS16. We have added a line to the legend of Figure 9 to draw the reader’s attention to both views.

6. While the 3D PDFs allow for user friendly exploration of the distinct structures present during development, the authors may consider creating a time lapse video for describing complex septation processes and valvular development. Particularly those events described in figure 8 and 9, the septation of the outflow tract both in the proximal muscular segments and distal vascular segments are unclear based on the parallel endocardial ridges and neural crest structures that are shown in the figures.

Our Figures have 2 aims: showing interesting features and suggesting good viewing points to inspect difficult topography. Figures 5 and 6 belong to the latter category. We can imagine that a movie can support the understanding of e.g. a Ca²⁺ or depolarization wave very effectively. We do wonder, however, if this also applies to static images. In our experience, one watches a rotating reconstruction only once. To really understand a complex feature like septation 3D-PDFs are much more instructive in our experience, because they allow the readers to manipulate the views themselves, with all the advantages of movement of structures and adding, making transparent, or removing one or more structures kept intact. We did add, however, a movie on how to use a 3D-PDF effectively, as we have received comments that readers experienced this as initially difficult.

Reviewer #3 (Remarks to the Author):

The authors have reconstructed 12 specimens from the Carnegie collection of histological sections, providing a 3D atlas of human embryonic development, with annotated segmentations of cardiac structures. Outstanding 3D-pdfs are provided, for an interactive visualization. 12 figures recapitulate the temporal progression of specific features. The text describes each anatomical stage and refers to mechanistic insights obtained in the experimental mouse model. Overall, this is a very careful, rather complete and well-referenced study, which will be useful for anatomists, developmental biologists, pediatric cardiologists, as well as students. I have minor comments to clarify aspects of the study.

1- How the structures have been segmented based on histological features is not fully clear. Segmentation is an interpretation, and it is important to show examples of how the contours superimpose on raw data.

We now provide the workflow of making our reconstructions in Supplemental Figure 17.

It is unclear why the arterial most part of the early heart is referred to as “outflow tract” in the text and 3D-pdfs of CS10-11, when the authors elsewhere (ex line 970) and in the Table acknowledge that the RV is added before the OFT to the tube. In the mouse, the early heart tube similar to that shown at CS10 was shown to contain LV and RV cells (Zaffran et al, 2004, ref 43).

We thank the reviewer for this critical comment. As stated by the reviewer, we consider the future right ventricle as the component of the second heart field that is first added downstream to the embryonic left ventricle. In our text we have referred to that structure as outflow tract, because we could not yet subdivide the cardiac lumen and muscular wall in a right ventricular and outflow-tract portion. In response to the comment of the reviewer, we have reread the literature on very early heart development. Most authors refer to the structure at this stage as the (embryonic) right ventricle and that the outflow tract in the strict sense is added to the embryonic right ventricle (e.g. Biben and Harvey, 1997, Zaffran et al., 2004, Le Garrec et al., 2017). We have modified the text, and PDF marker colors and legends accordingly.

2- I do not understand how Supplementary Figure 1 has been drawn. Which criteria were used for the inter-species comparison? Please illustrate the comparisons, for example using the HREM images of Mohun and Anderson. I am surprised to see a linear relationship: (1) Development doesn't have to be proportional between species; (2) In this graph, human stages are based on morphological features whereas mouse stages are based on time. To me CS9 (shown in Ref 9 or 24) to CS12 all correspond to successive looping stages at E8.5 in the mouse. Others have proposed a looser correlation (Krishnan et al., 2014).

Mouse development is, as the reviewer states, usually expressed in embryonic days, assuming that development starts at midnight of the overnight mating period. This practice, which probably arose to accommodate the use of multiple mouse embryos in a single experiment, contributes substantially to the mentioned variation in the existing staging systems. To prepare Supplemental Figure 1 (now Supplemental Figure 13), we have compared human developmental “Horizons” according to Streeter (1951; relabeled Carnegie stages by O’Rahilly (1971a) with mouse developmental stages according to Theiler (1972). The first step was, therefore, a comparison of morphological stages. Because the common use of embryonic days to stage mouse embryos, we have used Theiler’s corresponding days of development (ED7.5-14.5) in the text. The comparison between chick and human heart development is directly taken from (Kirby, 2007).

We opted for Theiler’s staging system after looking into many efforts to establish a staging system that relates human and murine (heart) development. In particular, we have compared Theiler (1972), Butler and Juurlink (1987), Buckingham et al. (2005) *plus* Krishnan et al. (2014) for early *and* late stages of heart development, respectively, Krishnan et al. (2014), and de Bakker et al. (2016). For each Carnegie stage, mouse age differed by as much as 1-2 days among these sources! Theiler (1972) and Buckingham et al. (2005) *plus* Krishnan et al. (2014) differed least. Eventually, we have opted for Theiler’s staging system, because his system is based on the morphological criteria also used to score Streeter’s Horizons, which, in turn, were incorporated virtually unchanged in O’Rahilly’s Carnegie stages (O’Rahilly and Muller, 1987). Finally, we also checked whether Lawson’s refinement of Theiler’s stages 9-12 (<http://www.emouseatlas.org>) affected the correlations, which was not the case. In the course of these comparisons we discovered, however, that we made a mistake. Streeter proposed his staging system very briefly in 1942 (Streeter, 1942), but only described stages

(“Horizons”) 11-23 in detail between 1942 and his death in 1951. Stages 0-10 were described in detail by O’Rahilly and Müller as “Carnegie stages” (O’Rahilly and Muller, 1987). Carnegie stages differ from Horizons in that Carnegie stages 8, 9, and 10 correspond best with Horizon VIII-IX, X/a, and X/b, respectively (numbering: Theiler), while Carnegie stage 11 is again comparable to Horizon XI. Theiler used the Streeter’s 1942 brief list of stages I-X, which does not include detailed descriptions of these stages. In the submitted version, we just copied the Horizons mentioned in Theiler’s book rather than converting them to the Carnegie system. We have corrected this error, but the effect is only that the correlation between Carnegie stages and mouse embryonic days in the period studied becomes tighter. Our graph does not address intra- and inter-litter variability, but needs to be looked at in the way most experiments are conducted: a mouse embryo at 8.5 dpc is considered to have developed to Theiler’s stage 12.

As the reviewer states development need not be proportional between species. Pertinent to this issue in the present context is the comparison of mouse and human development by Otis and Brent (1954). These investigators established that mouse development between ED9 and ED14 was 4 times faster than human development (also expressed in days), and that between ED13.5 and ED16.5, mouse development abruptly increased more than 3-fold to >13-fold faster than human development. The developmental relation between human and mouse embryos between ED9 and ED13.5 can be described, therefore, with a single straight line.

Otis & Brent’s starting point was ED8, so the early phase of heart development may obey a different relation. This caveat is relevant, because early heart development in mice is perceived as proceeding very rapidly, and embryonic days 7.5 to 8.5 have been subdivided to accommodate this rapid development (e.g. Biben and Harvey, 1997, Tyser et al., 2016, Le Garrec et al., 2017, Ivanovitch et al., 2017). For that reason, we have validated our graph for that period by comparing mouse and human heart development expressed in days since fertilization (Supplemental Figure 15). We have compared the shape of our early human heart reconstructions with those used in Le Garrec’s staging system for early mouse heart development (Le Garrec et al., 2017). If we correlate Le Garrec’s mouse data (expressed in days of development) with the proposed number of days of development of CS9-12 human embryos in O’Rahilly and Muller (2010), mouse development is also ~4-fold faster than human development during the early phase of heart development. We have discussed this issue in our Discussion (page 40) and in the legend to Supplemental Figure 15.

3- The quantitative analyses should be treated with a bit more caution, as based on single individuals. The legend should explain that one dot is one sample. The paragraph.

“differences between specimens could usually be explained as small differences in degree of individual development rather than deviation from the expected morphology” is not a conclusion that can be reached before quantitative parameters are measured on a batch of embryos. CS9 shown here is for example very different from CS9 shown in Ref 9 or 24.

We do agree, of course, that measurements on single embryos need to be treated with caution. We have included the sentence that one dot represents one sample in all Figures with a graph. It is, however, our experience that timelines of single measurements often produce highly significant correlations. Apparently, the difference between time points exceeds the variation in measurements of single time points considerably. We further disagree with the reviewer that the CS9 stages in de Bakker et al. (2016) and Sizarov et al. (2011) are very different from ours. In fact, this is a nice example: it is reasonably simple to sort and separate the hearts of these CS9 embryos from the CS10 embryo we have reconstructed, even though the CS10 embryo has developed only 3 more somites than the CS9 embryos. Embryos can qualify as CS9late or CS10early, of course. But even then, it is usually possible to assign such an embryo a borderline position. Another example is Figure 10A: although measurements per stage vary, it is fairly easy to draw smooth lines. Similarly, Figure 7, upper left panel, shows a Figure with more observations per stage. Even measurements obtained from embryos processed according to 2 different techniques (SEM & immunohistochemistry) yield similar data across time.

4- " We did not encounter major gaps in our description of the models ". Without diminishing the quality of their study, the authors should acknowledge that early heart development is barely covered, from the cardiac crescent to tube formation and to tube looping. These stages have been shown in the mouse to be very dynamic (Tyser et al 2016, Tyser et al 2021, and ref 43) and would require segmentation of more CS9-CS12 embryos to be fully caught.

We have only 4 examples of hearts between the formation of the heart tube and the completion of heart looping. And, of course, we agree that examples of additional intermediate stages would improve the quality of our description. The same comment applies to e.g. valve development, or the appearance of the “freestanding” infundibulum guarding the pulmonary root. We anticipate to add embryos of intermediate stages in the future. What we meant with the sentence that “we did not encounter major gaps in our description of the models” is that the transition of earlier configurations into later arrangements could be described without having to assume major steps in development that could not be supported by data. We have clarified the sentence (sub limitations of the study).

5- Another limitation of the approach based on histological sections, is that the z-resolution is poor, as well as the registration of sections.

This criticism is also correct. We would like to add, however, that more sections mean more voxels and the generation of unwieldy large data sets, which, in turn, hinders the quality of the reconstruction. Furthermore, with manual segmentation, the high X-Y resolution is only theoretical, since the accuracy of the contours that are produced, is much lower. Finally, the irregular surface of reconstructed structures, that arises due to histological artefacts and insufficiently accurate segmentation makes many reconstructions hard to interpret, even for the experienced investigator. And even when reconstructions are made with extreme care, their interpretation is often difficult without further processing (see e.g. the reconstruction of endocardial jelly in Faber et al. (2021)). To facilitate interpretation of the reconstructions, we use the Cinema4D program.

6- It is a pity that CS15 could not be reconstructed entirely. Could an alternative sample be found, like the one segmented in Ref 9 or from another embryo collection ?

We have decided to complete the CS15 heart and embryo after all. The reason to continue with this embryo is that many embryos of stages 15 and 16 that are recovered show moderate-to-severe venous congestion, suggesting that cardiac failure is a frequent cause of death in the 6th week of development. The reconstructed embryos in the present study were selected from the Carnegie collection, while many other embryos that we have recently described, were selected from the departmental collections in Leiden and Amsterdam. These embryos could not be used, therefore, to score prevalence of venous congestion. Recently, Prof Viebahn (Goettingen) provided us with access to Blechschmidt's collection of embryos. Among the 33 embryos that were scanned, 10 showed clear signs of venous congestion, with a clear maximum in prevalence at Carnegie stages 15 and 16 (Supplemental Figure 14). We have opted to complete the CS15 DREM embryo, because we were not involved in the assignment of a Carnegie stage to this embryo. Therefore, all our examples are based on embryos staged elsewhere. We have modified the pertinent text.

7- Anatomical descriptions are very useful. However, a caveat is to draw mechanistic conclusions which can instead only result from experimental conditions. See for example lines 170 or 570-571, which are confusing in this regard. In the text, the authors should make it clearer when they refer to previous mouse experimental data, compared to their own original anatomical observations. Example : "The first heart field, which is defined during CS8 ref 16" is incorrect. Do the authors mean "The first heart field, which has been shown in the mouse to arise at a stage equivalent to CS8 " ? See also line 195

The reviewer is right. We have gone over our text and modified many examples. Adhering very strictly to this rule produces awkward texts, however. We have ensured, therefore, that in all cases the reference to the source of the claim is present.

8- The statement “that the first heart field (..) gives rise primarily to the embryonic left ventricle” is wrong. Clonal data (ref 63) clearly show that it also contributes to the RV, the AVC and the atria. Ref 19, 20 are based on genetic tracing, which is not an appropriate method to trace the lineage of cells (Ref 36).

We thought we were sufficiently careful by using “primarily”. In this respect, we just follow the text used in many introductions or summaries to articles. Lescroart et al. (2018) e.g. state that “at E6.5, Mesp1+ cells mark left ventricle (LV) progenitors, whereas the right ventricle, atria, outflow, and inflow tracts and head muscles arise from Mesp1+ cells at E7.25, which correspond respectively to the first and second heart fields (FHF and SHF, respectively)”. We acknowledge, nonetheless, that our descriptions should be correct and have modified the text in the section on the CS9 embryos accordingly.

The comment of the reviewer made us realize that we should address the remodeling of the embryonic into the definitive ventricles more carefully. When ventricular septation starts, the free wall of the left ventricle, which initially consists of purely FHF cells, becomes “diluted” by Tbx2+ cells of the atrioventricular canal (Aanhaanen et al., 2009), while a sharp boundary develops between left and right side of the muscular ventricular septum due to a predominantly circumferential growth pattern (Meilhac et al., 2004, Zaffran et al., 2004, Devine et al., 2014). Furthermore, the myocardium of the proximal OFT becomes incorporated into the wall of the RV starting at CS17. In the human embryo, this argument is based on the remodeling of the OFT myocardium, but in chicks, this relocation of boundaries has been followed with markers (van den Hoff et al., 1999).

9- The statement that the heart starts beating at CS10 is surprising. In the mouse, cardiomyocytes beat as soon as they differentiate, and so already at the crescent stage (Tyser et al., 2006)

It was our aim to indicate at what stage the heart starts beating visibly. In response to the reviewers’ comments, we have reread some of the pertinent literature, including those by Tyser et al. (2016), and Tyser and Srinivas (2020). We do realize that the registration of more sensitive parameters, such sarcomeric assembly, and Ca²⁺ or transmembrane potential transients will precede visible contractions, and allow a more accurate attribution. We have modified our text as follows: Cardiomyocyte contractions first arise in the lateral regions of the cardiac crescents of mouse embryos as soon as sarcomeric assembly and Ca²⁺ transients can be demonstrated (Kobayashi et al., 2011, Tyser et al., 2016, Ivanovitch et al., 2017).

10- Since the text refers a lot to mouse data, it would be useful to have a paragraph discussing differences in the two species. I was surprised by (1) the massive jelly at CS9, (2) the lack of outflow cushions/ridges at CS13, whereas AVC cushions are segmented, (3) the extension of AVC cushions into the atrium. Are these human features?

Our team has studied heart development from the perspective of the human embryo and studies mouse heart development mainly to learn about mechanistic sequences. The features the reviewer mentions are, therefore, partly new to us. The amount of jelly seen at CS9 human hearts is seen in all human hearts of that stage (Davis, 1927), so certainly an established feature. Similarly, the epithelio-mesenchymal transition in, and the molding of the outflow ridges is occurring at CS14 in 25 of 45 embryos and is complete at CS15 (McBride et al., 1981), so the timing of that event is also normal. The extension of the atrioventricular cushions into the atria is also seen in mice and appears more a question of conventions in description.

11- “The terms proximal and distal refer to positions relative to the center of the heart” : isn't it rather referring to the sense of blood circulation ?

Proximal and distal do, indeed, refer to positions relative to the center of the heart. That language is common use in the description of the developing heart and other organs. With respect to the blood circulation, we use upstream and downstream (of the heart).

12- Please explain in the legend of the CS9 3D-pdf the green stripes. Please annotate the aorta as distinct from the pulmonary trunk in Figs 7-10, and the fishmouse in Fig. 7

The green stripes were meant to delineate the developing myocardium covering the jelly. We opted for this coding because sarcomeric proteins accumulate only gradually. We realize that that coding may be confusing and have modified it to show the myocardium separately.

We have not given the aorta and pulmonary trunk different color labels, because we already have (too) many labels. Instead, we have added to the legend that the aorta is recognizable by its “T-bar”, that is, its extrapericardial portion.

Typo:

Line 124 and 224: rightward

The reviewer is right. We were describing the leftward convex bend near the junction of the descending and ascending loop, but then also included that we were describing the cranial part. We have changed leftward to rightward (line 149 and legend to Figure 3).

Line 397 : is GIN2 equivalent to GIN ?

There were two cell lines expressing different monoclonal antibodies against an extract of the chicken nodose ganglion. The cell line we initially used was number 2 (Wessels et al., 1992). The GIN2 antibody recognizes the same antigen as Leu7 and Hnk1.

The link <https://www.prenatalorigins.org/virtual-human-embryo> doesn't work

Thanks for drawing our attention to this issue. We have removed the link.

References

- Aanhaanen WT, Brons JF, Dominguez JN, et al. (2009) The Tbx2+ primary myocardium of the atrioventricular canal forms the atrioventricular node and the base of the left ventricle. *Circ Res*, **104**, 1267-74.
- Anderson RH, Chaudhry B, Mohun TJ, et al. (2012) Normal and abnormal development of the intrapericardial arterial trunks in humans and mice. *Cardiovasc Res*, **95**, 108-15.
- Anderson RH, Webb S, Brown NA, Lamers W, Moorman A (2003) Development of the heart: (3) formation of the ventricular outflow tracts, arterial valves, and intrapericardial arterial trunks. *Heart*, **89**, 1110-8.
- Bartelings MM, Gittenberger-de Groot AC (1988) The arterial orifice level in the early human embryo. *Anat Embryol (Berl)*, **177**, 537-42.
- Biben C, Harvey RP (1997) Homeodomain factor Nkx2-5 controls left/right asymmetric expression of bHLH gene eHand during murine heart development. *Genes Dev*, **11**, 1357-69.
- Buckingham M, Meilhac S, Zaffran S (2005) Building the mammalian heart from two sources of myocardial cells. *Nat Rev Genet*, **6**, 826-35.
- Butler H, Juurlink BHJ (1987) *An atlas for staging mammalian and chick embryos.*, CRC Press, Boca Raton, FL.
- Chan SS, Shi X, Toyama A, et al. (2013) Mesp1 patterns mesoderm into cardiac, hematopoietic, or skeletal myogenic progenitors in a context-dependent manner. *Cell Stem Cell*, **12**, 587-601.
- Davis CL (1927) Development of the human heart from its first appearance to the stage found in embryos of twenty paired somites. *Contributions to embryology, Carnegie Institution*, **107**, 247-284.
- de Bakker BS, de Jong KH, Hagoort J, et al. (2016) An interactive three-dimensional digital atlas and quantitative database of human development. *Science*, **354**.
- Devine WP, Wythe JD, George M, Koshiba-Takeuchi K, Bruneau BG (2014) Early patterning and specification of cardiac progenitors in gastrulating mesoderm. *Elife*, **3**.
- Faber JW, Hagoort J, Moorman AFM, Christoffels VM, Jensen B (2021) Quantified growth of the human embryonic heart. *Biol Open*, **10**.
- Harmon AW, Nakano A (2013) Nkx2-5 lineage tracing visualizes the distribution of second heart field-derived aortic smooth muscle. *Genesis*, **51**, 862-9.
- Ivanovitch K, Temino S, Torres M (2017) Live imaging of heart tube development in mouse reveals alternating phases of cardiac differentiation and morphogenesis. *Elife*, **6**.
- Jin H, Wang H, Li J, et al. (2019) Differential contribution of the two waves of cardiac progenitors and their derivatives to aorta and pulmonary artery. *Dev Biol*, **450**, 82-89.
- Kirby ML (2007) *Cardiac development*, Oxford University Press, New York.
- Knaapen MW, Vrolijk BC, Wenink AC (1995) Growth of the myocardial volumes of the individual cardiac segments in the rat embryo. *Anat Rec*, **243**, 93-100.
- Kobayashi T, Maeda S, Ichise N, et al. (2011) The beginning of the calcium transient in rat embryonic heart. *J Physiol Sci*, **61**, 141-9.
- Krishnan A, Samtani R, Dhanantwari P, et al. (2014) A detailed comparison of mouse and human cardiac development. *Pediatr Res*, **76**, 500-7.
- Le Garrec JF, Dominguez JN, Desgrange A, et al. (2017) A predictive model of asymmetric morphogenesis from 3D reconstructions of mouse heart looping dynamics. *Elife*, **6**.
- Lescroart F, Wang X, Lin X, et al. (2018) Defining the earliest step of cardiovascular lineage segregation by single-cell RNA-seq. *Science*, **359**, 1177-1181.

- Li D, Angermeier A, Wang J (2019) Planar cell polarity signaling regulates polarized second heart field morphogenesis to promote both arterial and venous pole septation. *Development*, **146**.
- Liang S, Li HC, Wang YX, et al. (2014) Pulmonary endoderm, second heart field and the morphogenesis of distal outflow tract in mouse embryonic heart. *Dev Growth Differ*, **56**, 276-92.
- McBride RE, Moore GW, Hutchins GM (1981) Development of the outflow tract and closure of the interventricular septum in the normal human heart. *Am J Anat*, **160**, 309-31.
- Meilhac SM, Esner M, Kerszberg M, Moss JE, Buckingham ME (2004) Oriented clonal cell growth in the developing mouse myocardium underlies cardiac morphogenesis. *J Cell Biol*, **164**, 97-109.
- O'Rahilly R (1971) The timing and sequence of events in human cardiogenesis. *Acta Anat (Basel)*, **79**, 70-5.
- O'Rahilly R, Muller F (1987) *Developmental stages in human embryos, including a revision of Streeter's "horizons" and a survey of the Carnegie Collection*.
- O'Rahilly R, Muller F (2010) Developmental stages in human embryos: revised and new measurements. *Cells Tissues Organs*, **192**, 73-84.
- Okamoto N, Akimoto N, Satow Y, Hidaka N, Miyabara S (1981) *Role of cell death on conal ridges of developing human heart*, Raven Press, New York.
- Orts-Llorca F, Puerta Fonolla J, Sobrado J (1982) The formation, septation and fate of the truncus arteriosus in man. *J Anat*, **134**, 41-56.
- Otis EM, Brent R (1954) Equivalent ages in mouse and human embryos. *Anat Rec*, **54**, 33-63.
- Sawada H, Rateri DL, Moorleghen JJ, Majesky MW, Daugherty A (2017) Smooth muscle cells derived from second heart field and cardiac neural crest reside in spatially distinct domains in the media of the ascending aorta-brief report. *Arterioscler Thromb Vasc Biol*, **37**, 1722-1726.
- Sizarov A, Ya J, de Boer BA, Lamers WH, Christoffels VM, Moorman AF (2011) Formation of the building plan of the human heart: morphogenesis, growth, and differentiation. *Circulation*, **123**, 1125-35.
- Streeter GL (1942) Developmental horizons in human embryos. . *Contr Embryol Carneg Instn*, **30**, 213-230.
- Streeter GL (1951) Developmental horizons in human embryos. In *Contrib Embryol*, (eds Heuser CH, Corner GW), pp. 165-196. Washington, D.C.: Carnegie Inst Wash.
- Theiler K (1972) *The house mouse. Development and normal stages from fertilization to 4 weeks of age.*, Springer-Verlag, Berlin.
- Thompson RP, Sumida H, Abercrombie V, Satow Y, Fitzharris TP, Okamoto N (1985) Morphogenesis of human cardiac outflow. *Anat Rec*, **213**, 578-86, 538-9.
- Tyser RC, Miranda AM, Chen CM, Davidson SM, Srinivas S, Riley PR (2016) Calcium handling precedes cardiac differentiation to initiate the first heartbeat. *Elife*, **5**.
- Tyser RSV, Srinivas S (2020) The first heartbeat—origin of cardiac contractile activity. *Cold Spring Harb Perspect Biol* **12**, 12.
- van den Hoff MJ, Moorman AF, Ruijter JM, et al. (1999) Myocardialization of the cardiac outflow tract. *Dev Biol*, **212**, 477-90.
- Webb S, Qayyum SR, Anderson RH, Lamers WH, Richardson MK (2003) Septation and separation within the outflow tract of the developing heart. *J Anat*, **202**, 327-42.
- Wessels A, Vermeulen JL, Verbeek FJ, et al. (1992) Spatial distribution of "tissue-specific" antigens in the developing human heart and skeletal muscle. III. An immunohistochemical analysis of the distribution of the neural tissue antigen GIN2 in the embryonic heart; implications for the development of the atrioventricular conduction system. *Anat Rec*, **232**, 97-111.
- Yang YP, Li HR, Cao XM, Wang QX, Qiao CJ, Ya J (2013) Second heart field and the development of the outflow tract in human embryonic heart. *Dev Growth Differ*, **55**, 359-67.
- Zaffran S, Kelly RG, Meilhac SM, Buckingham ME, Brown NA (2004) Right ventricular myocardium derives from the anterior heart field. *Circ Res*, **95**, 261-8.
- Zhou Z, Wang J, Guo C, et al. (2017) Temporally distinct Six2-positive second heart field progenitors regulate mammalian heart development and disease. *Cell Rep*, **18**, 1019-1032.

REVIEWERS' COMMENTS:

Reviewer #2 (Remarks to the Author):

The authors have provided adequate responses to the reviewer comments, and I thank them for incorporating clarifications into the descriptions of the early heart fields and gene markers used for their identifications. The authors have added significant discussion points that I believe has increased the overall impact of their manuscript for the field of cardiac development by conducting an in-depth comparison between murine and human model of embryonic heart development.

In addition to addressing the comparisons between species, the authors have significantly improved the clarity of the figures with the changes to color palette used for visualization. The changes made to create greater distinction between structures with increased weight of the contour lines (particularly in Figure 5) has helped to gain better understanding of the septations events described by the authors. I have also found that the addition of visual cues such as the line in Figure 9 has improved my understanding of the separation between the aortic and pulmonary channels.

Overall, I believe that this manuscript provides an excellent resource and analysis of human cardiac development. The remarkable similarity between human and murine models of cardiac development is reassuring in terms of modeling important events during early cardiac development. One issue that may be worth discussing is whether current methods for staging human embryos are likely underestimating the age of the specimens. In this study, the authors used well described staging methods to estimate the age post fertilization. The lack of adequate staging appears to be a critical issue due to the rapid developmental changes that occur during the early stages of development as the authors have pointed out. In conclusion, I believe that the manuscript as submitted is ready for publication.

Reviewer #3 (Remarks to the Author):

The authors have well addressed reviewer comments. I fully recommend publication of this fabulous resource and careful analysis of embryonic human heart development.

Answer to reviewers.

Reviewer #2:

The authors have provided adequate responses to the reviewer comments, and I thank them for incorporating clarifications into the descriptions of the early heart fields and gene markers used for their identifications. The authors have added significant discussion points that I believe has increased the overall impact of their manuscript for the field of cardiac development by conducting an in-depth comparison between murine and human model of embryonic heart development.

In addition to addressing the comparisons between species, the authors have significantly improved the clarity of the figures with the changes to color palette used for visualization. The changes made to create greater distinction between structures with increased weight of the contour lines (particularly in Figure 5) has helped to gain better understanding of the septations events described by the authors. I have also found that the addition of visual cues such as the line in Figure 9 has improved my understanding of the separation between the aortic and pulmonary channels.

Overall, I believe that this manuscript provides an excellent resource and analysis of human cardiac development. The remarkable similarity between human and murine models of cardiac development is reassuring in terms of modeling important events during early cardiac development. One issue that may be worth discussing is whether current methods for staging human embryos are likely underestimating the age of the specimens. In this study, the authors used well described staging methods to estimate the age post fertilization. The lack of adequate staging appears to be a critical issue due to the rapid developmental changes that occur during the early stages of development as the authors have pointed out. In conclusion, I believe that the manuscript as submitted is ready for publication.

We thank the reviewer for his/her positive evaluation. The issue whether the post-fertilization age of the embryo and its stage of development correlate is obviously of critical importance when comparing human development with a that of experimental animals, such as mice. As we have explained, we opted on purpose for the Carnegie staging system, as its stages 11-23 as described by Streeter (1, 2) were used by Theiler to develop a staging system for mice (3). Stages 9 (4) and 10 (5) were added later. Theiler does not state whether he consulted the descriptions of stages 9 and 10. For the stages studied, our approach yields a straight line between human and mouse age post fertilization, with ~4 human days being equivalent to ~1 mouse day (see our first answer to the reviewer).

The question then boils down to whether the estimate of the postfertilization age is correct. The correlation in rate of development between humans and mice that we observed can be ascribed at least in part to the revised post-fertilization ages described by O'Rahilly and Müller in 2010 (6). In this revision, which is based on an integration of ultrasound data, the embryos belonging to Carnegie stages 6-16, and in particular those belonging to Carnegie stages 8-15, were assigned a ~4 days younger age. If, in addition, Carnegie stages 5a, 5b, and 5c would be established as separate stages, a smooth, more or less linear correlation between

Carnegie stage and post-fertilization age of human development evolves. Such an intervention would require, however, a reassignment of developmental stages 6-23, which may be a step too far for many.

Shifting the observed correlation between stage and age towards younger ages runs counter the opinion of the reviewer that we underestimate age. Clearly, the age axis in the correlation graph is less strongly anchored in data than the stage axis. From our point of view, this issue can best be addressed by longitudinal high-resolution ultrasound observations of healthy pregnancies.

Reviewer #3:

The authors have well addressed reviewer comments. I fully recommend publication of this fabulous resource and careful analysis of embryonic human heart development.

We thank the reviewer for his/her positive comment.

1. Streeter GL. Developmental horizons in human embryos. Contrib Embryol, Carnegie Instn Wash. 1942;30:213-30.
2. Heuser CH, Corner GW. Developmental horizons in human embryos. Contrib Embryol, Carnegie Instn Wash. 1951;34:165-96.
3. Theiler K. The house mouse. Development and normal stages from fertilization to 4 weeks of age. Berlin: Springer-Verlag; 1972. 168 p.
4. O'Rahilly R. Developmental stages in human embryos, including a survey of the Carnegie collection. Part A: Embryos of the first three weeks (stages 1 to 9). Contrib Embryol, Carnegie Instn Wash Pub 631; 1973.
5. Heuser CH, Corner GW. Developmental horizons in human embryos. Description of age group X, 4 to 10 somites. Carnegie Inst Wash Pub 244, Contrib Embryol. 1957;36:29-39.
6. O'Rahilly R, Muller F. Developmental stages in human embryos: revised and new measurements. Cells Tissues Organs. 2010;192(2):73-84.